# Usp9x regulates Ets-1 ubiquitination and stability to control NRAS expression and tumorigenicity in melanoma

Harish Potu[1], Luke F. Peterson[1], Malathi Kandarpa[1], Anupama Pal[1], Hanshi Sun[2], Alison Durham[3], Paul W. Harms[4], Peter C. Hollenhorst[5], Ugur Eskiocak[6,†], Moshe Talpaz[1] & Nicholas J. Donato[7]

ETS transcription factors are commonly deregulated in cancer by chromosomal translocation, overexpression or post-translational modification to induce gene expression programs essential in tumorigenicity. Targeted destruction of these proteins may have therapeutic impact. Here we report that Ets-1 destruction is regulated by the deubiquitinating enzyme, Usp9x, and has major impact on the tumorigenic program of metastatic melanoma. Ets-1 deubiquitination blocks its proteasomal destruction and enhances tumorigenicity, which could be reversed by Usp9x knockdown or inhibition. Usp9x and Ets-1 levels are coincidently elevated in melanoma with highest levels detected in metastatic tumours versus normal skin or benign skin lesions. Notably, Ets-1 is induced by BRAF or MEK kinase inhibition, resulting in increased NRAS expression, which could be blocked by inactivation of Usp9x and therapeutic combination of Usp9x and MEK inhibitor fully suppressed melanoma growth. Thus, Usp9x modulates the Ets-1/NRAS regulatory network and may have biologic and therapeutic implications.

[1] Department of Internal Medicine/Division of Hematology/Oncology, University of Michigan School of Medicine and Comprehensive Cancer Center, Ann Arbor, Michigan 48109, USA. [2] Department of Periodontics and Oral Medicine, University of Michigan School of Dentistry, Ann Arbor, Michigan 48109, USA. [3] Department of Dermatology, University of Michigan School of Medicine, Ann Arbor, Michigan 48109, USA. [4] Departments of Pathology and Dermatology, University of Michigan School of Medicine, Ann Arbor, Michigan 48109, USA. [5] Department of Biochemistry and Molecular Biology, Medical Sciences Program, Indiana University Bloomington, 1001 Third St, Bloomington, Indiana 47405, USA. [6] Children's Research Institute and Department of Pediatrics, Howard Hughes Medical Institute, University of Texas Southwestern Medical Center, Dallas, Texas 75390, USA. [7] Department of Pharmacology, University of Michigan School of Medicine, Ann Arbor, Michigan 48109, USA. † Present address: Compass Therapeutics, 450 Kendall Street, Cambridge, Massachusetts 02142, USA. Correspondence and requests for materials should be addressed to N.J.D. (email: ndonato@umich.edu).

Recent progress has been made in targeting pathways activated by mutations in metastatic melanoma, and these advances have led to major improvements in patient treatment and survival[1]. However, many biological and clinical characteristics of melanoma are still unknown and current targeted therapies (BRAF and/or MEK inhibitors) are only effective in a subset of patients and typically for a limited duration (4–12 months)[2]. Combination kinase inhibitor therapy can circumvent or delay resistance and reactivation of immune responsiveness has shown some promising results. However, these therapies are only effective in 30–40% of patients and serious side effects (that is, auto-immunity) limit sustained clinical benefit, highlighting the need for novel strategies that could add to existing therapies[3]. Adjoined to that need, is the lack of understanding of some of the basic biology of melanoma, particularly what underlies the progression to metastatic disease after driver mutations are in place. Some recent studies have provided insight and have suggested that age, environmental factors and diet may underlie the transition[1,4,5].

The ubiquitin-proteasome system (UPS) has received considerable attention as a source of new drug targets because of the clinical success of 20S proteasome inhibitors in specific cancers. The UPS has multiple components that are considered targetable[6,7]. Among them are deubiquitinases (DUBs): enzymes that mediate removal of ubiquitin monomers or polymers from target proteins, and are major regulators of the UPS. Many DUBs demonstrate specificity for proteins involved in disease-associated pathways and are deregulated in disease by mutations, altered expression or post-translational modification[8–10]. Ubiquitin specific peptidase 9, X-linked (Usp9x), also known as FAF; FAM; DFFRX and MRX99, is a high MW DUB that has been shown to be over-expressed in several cancers, but can have both positive and negative impact on tumorigenicity, depending on the cancer type and disease model studied[11–16]. Usp9x deubiquitinates proteins essential in tumour cell signalling and survival, protecting some of them from proteasomal destruction[14,15,17].

The ETS (E26 transformation-specific or E-twenty-six; based on the gene transduced by the leukaemia virus, E26) transcription factor family is composed of 28 members, which recognize a DNA binding sequence minimally consisting of GGA(A/T)[18–20]. Specific members of this highly conserved family are frequently activated by chromosomal translocation, overexpression and stabilization (by altered ubiquitination) and are essential in tumorigenesis[21]. For example, FLI1 and ERG are overexpressed in Ewing sarcoma and prostate cancer as a consequence of chromosomal translocation and are key drivers of these malignancies[22,23]. Ets-1, and other family members, are overexpressed and regulated (positively and negatively) by phosphorylation, sumoylation and ubiquitination associated with specific signalling events[24–27]. Phosphorylation of specific ETS proteins mediated by an aberrant RAS/RAF/MEK/ERK signalling pathway provides one mechanism for promoting gene expression essential in driving the cancer phenotype and dominant negative versions of ETS genes can block oncogenic RAS/ERK tumorigenicity[19,28]. Ets-1 overexpression has been documented in many invasive and metastatic cancers, including breast, lung, colon, pancreatic and thyroid cancer[25,29–34], where Ets-1 drives gene expression associated with cellular differentiation, migration, proliferation, survival and angiogenesis. Members of the ETS transcription factor family are considered excellent therapeutic targets but most targeting approaches have failed[35].

This report provides evidence of an essential role for Usp9x in melanoma because of its regulation of Ets-1 protein levels. Through Usp9x-mediated, site-specific deubiquitination, Ets-1 proteasomal destruction is inhibited, resulting in Ets-1 accumulation and increased melanoma tumorigenicity, which could be blocked by inhibition of Usp9x activity or knockdown of Ets-1. We also determined that Ets-1 expression was negatively regulated by BRAF and/or MEK kinase activity and inhibition of this pathway increased Ets-1 expression to increase NRAS levels by activating the NRAS promoter. Since NRAS mutations are common (15–20%) in melanoma patients (and other cancers including multiple myeloma, lymphoma, lung, thyroid and colorectal cancer[36]) and its continual expression is essential for NRAS mutant melanoma cell growth and survival[37,38], NRAS mutant tumours were highly dependent on Usp9x. Thus, we provide evidence that Usp9x plays an important role in Ets-1 regulation and melanoma tumorigenicity, in part through NRAS transcription which may be of particular importance in tumours driven by NRAS mutation.

## Results

**Usp9x is required for *in vivo* melanoma growth.** We and others previously described Usp9x activity and expression in melanoma[10,39] and sought to define its role in primary and metastatic disease. Initially, we depleted Usp9x using a previously characterized shRNA knockdown (KD) vector[40] in three melanoma cell lines with distinct driver mutations (BRAF mutant: SK-Mel28, A375; NRAS mutant: SK-Mel147) and metastatic efficiencies (highly metastatic: A375, SK-Mel147) and compared biological effects to control cells. Usp9x knockdown (KD) modestly reduced the steady-state level of the anti-apoptotic protein Mcl-1 (a previously defined Usp9x substrate[14]), activated caspase cleavage (Fig. 1a) and reduced tumour growth under standard monolayer growth conditions (2D). However, Usp9x KD significantly impaired 3D melanoma growth, which is a better discriminator of the malignant and benign phenotype[41,42] (Fig. 1b,c). Usp9x depletion blocked expansive tumour growth in matrigel, particularly in tumours with NRAS mutations (Fig. 1c,d). To assess clinical relevance, we examined melanoma chemosensitivity to our recently described small molecule Usp9x inhibitor (G9)[39,43] and detected moderately greater sensitivity in NRAS versus BRAF mutant lines (Fig. 1e). Tumour cells grown in 3D had higher levels of Usp9x activity/expression than those measured in 2D cultures (confirmed in additional cell lines—Supplementary Fig. 1a) and G9 inhibited Usp9x activity in cells from either culture condition (Fig. 1f). Both Usp9x KD and G9 blocked anchorage-independent melanoma growth (Fig. 1g) and G9 dose-dependently inhibited melanoma growth in matrigel (Fig. 1h), with nM sensitivity against NRAS mutant cells (SK-Mel103; $IC_{50} \sim 300$ nM), suggesting that Usp9x plays a role in tumour expansion, particularly in tumours with an NRAS mutation.

To further elucidate the role of Usp9x in melanoma and examine the sensitivity of NRAS mutant tumours to Usp9x KD and inhibition, we first assessed the effects of Usp9x KD on specific RAS proteins in highly metastatic NRAS and BRAF mutant melanomas. Usp9x KD reduced NRAS protein levels in both NRAS and BRAF mutant cells with little to no effect on HRAS or KRAS expression (Fig. 2a). Previous studies demonstrated that continual expression of mutant NRAS was essential for NRAS mutant melanoma survival[37,44], and we confirmed that dependence in NRAS KD studies (Supplementary Fig. 1b). Usp9x KD suppressed NRAS, but not KRAS gene expression (Fig. 2b). Thus, Usp9x-mediated regulation of NRAS expression in melanoma, particulalrly in NRAS mutant cells, may partly underly their dependence on Usp9x for continual expansion and survival. However, Usp9x may alter other components within the RAS signalling pathway as we detected a paradoxical increase in ERK activation in Usp9x KD cells.

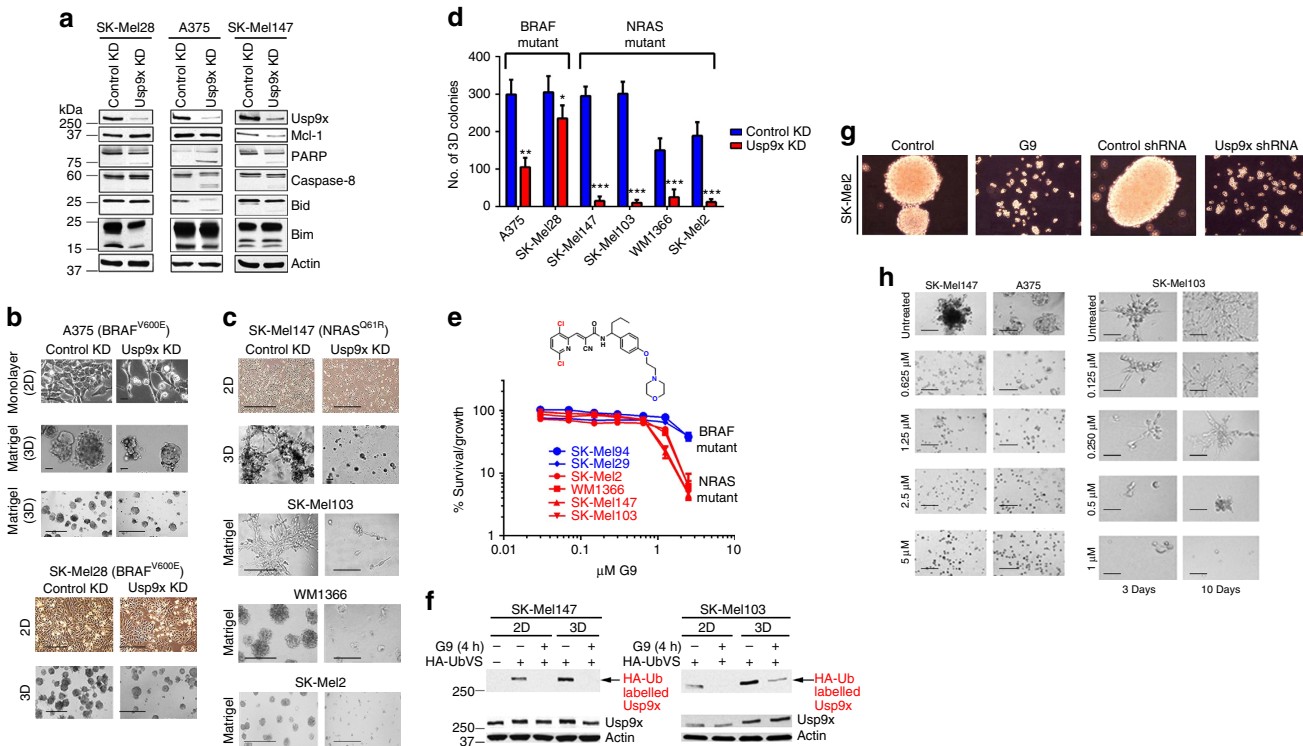

**Figure 1 | Effect of Usp9x KD and DUB inhibitor (G9) on the growth and expansion of melanoma cells.** (**a**) Immunoblot for the protein indicated in control and Usp9x KD (shRNA) melanoma cell lines. (**b**) Phase contrast images of BRAF mutant cells with or without Usp9x KD, grown in monolayer (2D—top) and matrigel (3D—bottom panels) for 7 days. Scale bars, 500 μm. (**c**) Phase contrast images of NRAS mutant cells with or without Usp9x KD, grown in 2D and 3D. Scale bars, 500 μm. (**d**) Quantification of colony growth in BRAF and NRAS mutant cells with and without Usp9x KD 7 days after plating. (**e**) Cell growth (by MTT) of NRAS mutant (SK-Mel2, WM1366, SK-Mel147, SK-Mel103) and BRAF mutant (SK-Mel94, SK-Mel29) cells treated with G9 at the indicated concentrations. The chemical structure of G9 (EOAI3401243) is shown. (**f**) DUB activity by HA-UbVS labelling in NRAS-mutant melanoma cells grown in 2D (monolayer) or 3D (agarose) and treated with G9 (5 μM, 4 h); HA-UbVS-labeled Usp9x is noted (top); Usp9x protein levels (bottom). (**g**) Phase contrast images of SK-Mel2 melanoma cells on agarose treated with or without 1 μM G9 for 3 days (left), and phase contrast images of control or Usp9x KD SK-Mel2 melanoma cells grown on agarose 3 days (right). (**h**) Phase contrast images of NRAS mutant (SK-Mel147) and BRAF mutant (A375) melanoma cells treated with G9 on matrigel for 3 days (left) and phase contrast images of NRAS mutant (SK-Mel103) melanoma cells treated with low dose of G9 (0–1 μM) on matrigel for 3 (left) or 10 days (right). Scale bars, 100 μm.

To determine the *in vivo* relevance of Usp9x in tumour expansion of NRAS mutant cells, equal numbers of viable control KD and Usp9x KD SK-Mel147 cells were transplanted into NSG mice and tumour growth was monitored over a 6-week interval. As shown in Fig. 2c, only one animal (of 3) had detectable tumour (shown) in mice injected with Usp9x KD cells, while control tumours grew to maximal burden in all 3 animals. We next enforced expression of Usp9x in HEK293T and SK-Mel29 cells (with low endogenous Usp9x expression) and detected upregulation of NRAS (Fig. 2d). Control and Usp9x-over-expressing SK-Mel29 cells were transplanted into NSG mice, and tumour growth was monitored in control and G9-treated mice (15 mg kg$^{-1}$, ip, QOD; begun after tumour was measurable) (Fig. 2e). Usp9x enforced expression increased tumour expansion by >2-fold over controls (red versus blue lines) and growth of Usp9x-overexpressing tumours could be blocked by *in vivo* G9 treatment (red versus green line). These results suggest that Usp9x enhances NRAS expression and *in vivo* tumour growth, which could be blocked by Usp9x depletion or inhibition.

**Usp9x modulates the melanoma ubiquitylome.** Analysis of Usp9x pulldowns failed to detect direct NRAS association or alterations in NRAS ubiquitination in Usp9x deficient or over-expressing cells. Therefore, we conducted an unbiased assessment of Usp9x-regulated ubiquitination in NRAS mutant melanoma to define potential targets and pathways that could mediate NRAS

regulation. The ubiquitylome induced by Usp9x KD or short-term G9 treatment (6 h) was compared with control cells (Supplementary Fig. 2a). Lysates from control, Usp9x KD and G9-treated SK-Mel147 cells were subjected to trypsinization and ubiquitin-remnant recovery[45,46]. Recovered Ub-peptides were identified following LC/MS/MS analysis and assignment of the spectral data. Multiple proteins were differentially ubiquitinated in Usp9x KD and G9-treated cells compared with controls (Fig. 3a), with predictive changes at specific amino acids (Supplementary Data 1 and 2). Positive and negative changes were noted and ~40% of the defined ubiquitylome was common to both Usp9x KD and G9-treated cells. Heat maps (Supplementary Fig. 2b; Supplementary Data 1–7) were constructed from two independent analyses, which suggested that Usp9x controls a broad range of ubiquitinated targets, with some previously identified as Usp9x substrates by other approaches[17]. Usp9x affected ubiquitination of multiple proteins within the UPS, including 11 DUBs, as noted in prior publications[43]. Identified targets were contributors to multiple pathways, with gene expression events being most prominent (REACTOME.org; Supplementary Fig. 2c; Supplementary Data 8).

To identify Usp9x targets with NRAS regulatory potential, we performed cluster analysis and screened for proteins within the Usp9x ubiquitylome with the following characteristics: (1) known effectors of the Ras pathway, (2) negative regulators of signal transduction and/or (3) transcription factors. We also searched

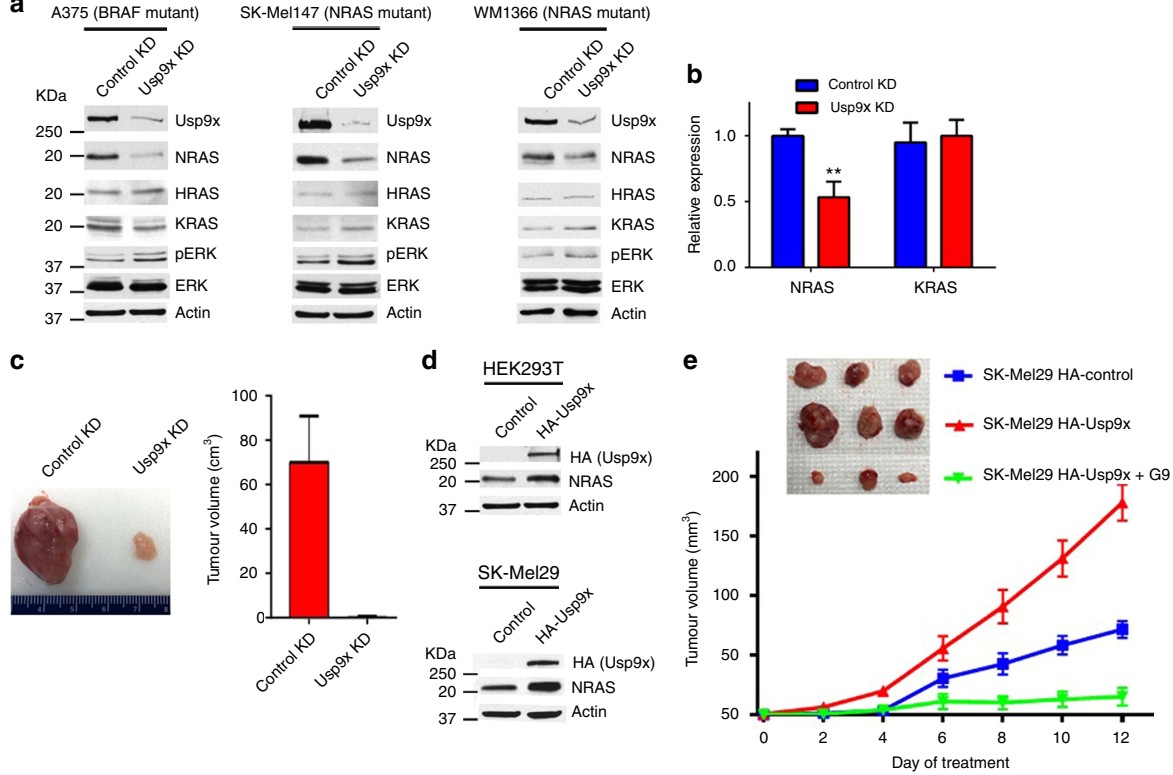

**Figure 2 | Usp9x regulates NRAS levels and is required for 3D growth.** (**a**) Immunoblot of RAS proteins and pERK in BRAF and NRAS mutant melanoma cells with and without Usp9x KD. (**b**) NRAS and KRAS gene expression in control and Usp9x KD SK-Mel147 cells by RT-PCR. (**c**) Tumour size in xenograft mice 6 weeks after injection with control ($N = 3$) or Usp9x ($N = 3$) KD SK-Mel147 cells. (**d**) Immunoblot for NRAS in 293T (top) or SK-Mel29 (bottom) control or Usp9x-overexpressing (HA-Usp9x) cells. Actin served as loading control. (**e**) Tumour volume in NSG mice injected subcutaneously with SK-Mel29 cells expressing HA-Control or HA-Usp9x. Mice were treated with vehicle (red, $N = 3$; blue, $N = 3$) or G9 (green, $N = 3$). At day 12 of treatment, tumours were excised and photographed (top).

the ubiquitylome for proteins known to interact with Usp9x or belonging to a protein family with a domain recognized by Usp9x. Specific ETS proteins emerged as possible contributors as several members have an essential role in tumorigenicity and embryonic development[19,20,24,33]. Ets-1 is both responsive to and a target of the RAS/MEK/ERK signalling pathway[24], and other members of the ETS family (that is, ERG, FLI1, FEV) have been shown to associate with and be deubiquitinated by Usp9x (ERG)[15]. Ub-remnant analysis indicated that both Usp9x KD and inhibition of activity with G9 increased Ets-1 (and its isoform), Ets-2, ETV2 and/or GABPα ubiquitination specifically within their ETS domain (K388 in Ets-1), a domain previously shown to be recognized by Usp9x (Fig. 3b)[15]. Since assignment is based on peptide sequence, we assessed lysates for changes to specific ETS proteins and solely detected significant reduction in Ets-1 in Usp9x KD cells (Fig. 3c) and we confirmed that Ets-1 is susceptible to proteasomal degradation (Supplementary Fig. 2d). Association between endogenous Usp9x and Ets-1 was detected by pulldown and immunoblotting (Fig. 3d). The active site Cys (C1566) of Usp9x was required for optimal Ets-1 binding in co-expression experiments (Fig. 3e), and the central domain of Usp9x, upstream from the catalytic site, was the primary site of Ets-1 interaction (Supplementary Fig. 2e). We determined that Ets-1 is primarily ubiquitinated with K63-linked polymers (Supplementary Fig. 2f), and Ets-1 reduction by Usp9x KD was blocked by 20S proteasome inhibition, indicating Ets-1 degradation is proteasome dependent[27] (Fig. 3f). Both Usp9x KD and G9 treatment increased Ets-1 ubiquitin content (Fig. 3g).

To assess the importance of the K388 ubiquitination site on Ets-1, we mutated it (K388R, K388A) and detected reduced Ets-1 ubiquitination compared with wild-type protein, indicating K388 serves as a site for ubiquitination (Fig. 4a). Enforced expression of Usp9x reduced recovery of ubiquitinated Ets-1 (Fig. 4b). We also expressed wild-type (WT) HA-Ets-1 and K388R mutant protein in SK-Mel29 cells and detected increased stability (longer half-life) of the mutant protein (Fig. 4c,d), indicating that K388 ubiquitination/deubiquitination plays a role in Ets-1 stability. To determine whether this site affects Ets-1 tumorigenic activity, mutant Ets-1 (K388R) was expressed in melanoma with low endogenous Ets-1 expression (SK-Mel29; Fig. 4e), and tumorigenic activity was assessed by monitoring colony formation (Fig. 4f) or plating on matrigel (Fig. 4g). Expression of the Ets-1 mutant was diminished (1.9-fold) when compared with the WT protein in melanoma, but equivalent expression was achievable in HEK293T cells (Supplementary Fig. 2g). Differential expression of the mutant protein may be because of expression of distinct E2/E3 enzymes in these cell types. Expression of both WT and mutant Ets-1 increased colony number and 3D growth of melanoma; however, after normalizing for expression levels, the K388R mutation conferred greater tumorigenicity compared with overexpression of the WT protein (Fig. 4h).

**Coincident Usp9x, Ets-1 and NRAS expression in melanoma.** To further investigate Ets-1 function in melanoma, Ets-1 expression was modulated in SK-Mel29 cells, and NRAS expression, colony formation and 3D growth were assessed. Ets-1 overexpression increased NRAS levels and colony formation (Supplementary Fig. 3a-left and Supplementary Fig. 3b), while Ets-1 KD reduced NRAS levels and blocked long-term survival of

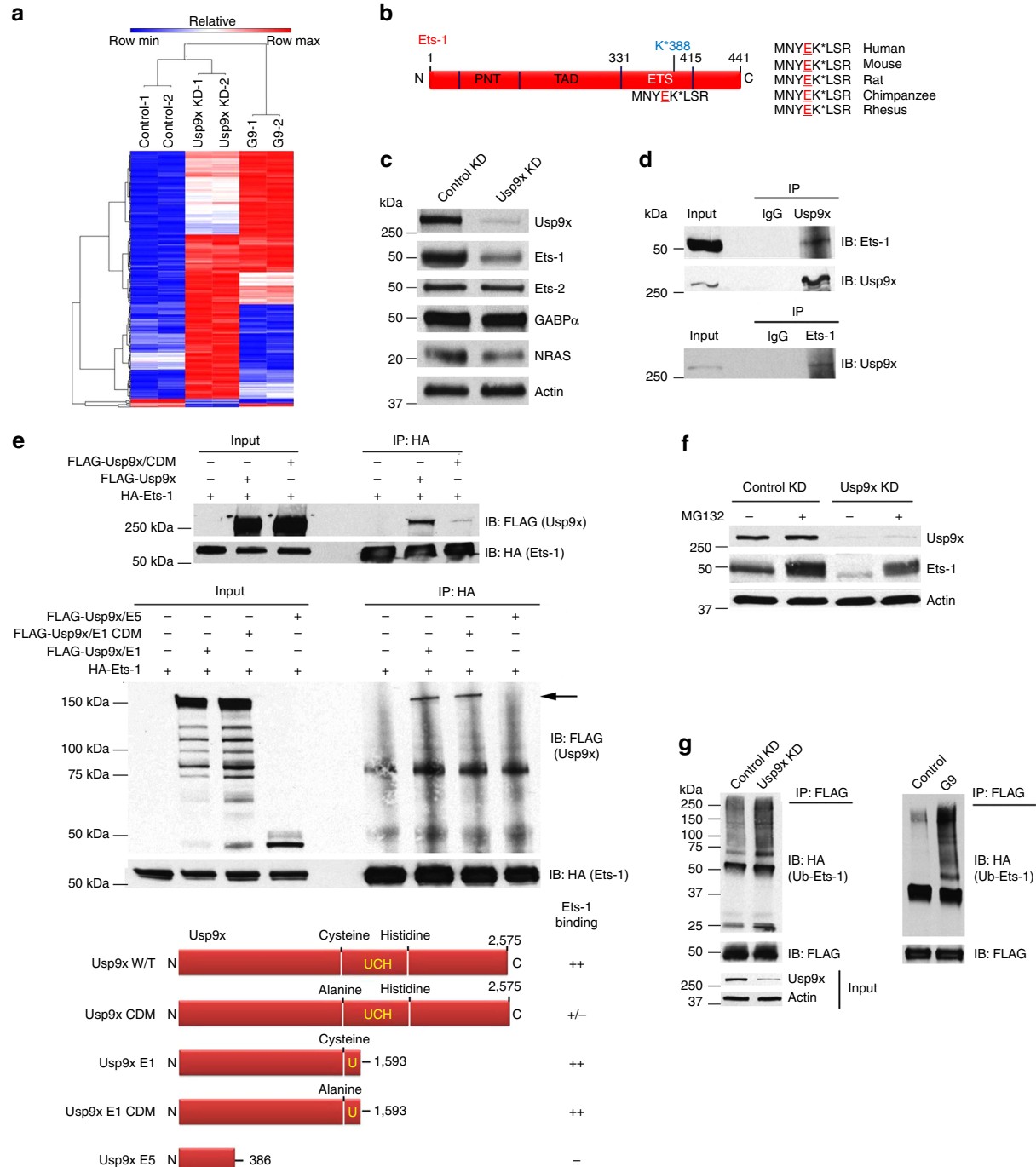

**Figure 3 | Usp9x deubiquitinates Ets-1 and regulates its degradation. (a)** Heat maps of differentially ubiquitinated proteins. NRAS mutant SK-Mel147 cells were exposed to control and Usp9x KD or G9 treatment as noted. The number of unique peptides and proteins reproducibly detected is shown. **(b)** Schematic diagram of the human Ets-1 protein showing the PNT (pointed domain, aa 53–136), TAD (transactivation domain, aa 137–242) and ETS domains. The putative site of ubiquitination (MNYEK*LSR) in human Ets-1 is shown and is conserved in mammalian species (right). **(c)** Immunoblot of ETS family proteins and NRAS in NRAS mutant melanoma cells with and without Usp9x KD. Actin served as a loading control. **(d)** Reciprocal immunoprecipitation of Usp9x and Ets-1 with endogenous Ets-1 and Usp9x in NRAS mutant SK-Mel2 cells. Immunoblotting was performed to detect Ets-1 or Usp9x in pulldowns and a portion of the input sample. **(e)** Top—Ectopically expressed FLAG-Usp9x (full-length) or FLAG-Usp9x-CDM (catalytic domain mutant, C1566A) was co-expressed with HA-Ets-1 in HEK293T cells. HA (Ets-1) immunoprecipitation was followed by immunoblotting of FLAG (Usp9x—top) or HA (Ets-1—bottom). Input lysate was also immunoblotted. Center—Ectopically expressed FLAG-Usp9x deletion constructs (FLAG-Usp9x E1, FLAG-Usp9x E1/CDM (catalytic domain mutant—C1566A), FLAG-Usp9x E5 (C-terminal deletion)) (illustrated in the bottom panel) were co-expressed with HA-Ets-1 in HEK293T cells. HA (Ets-1) immunoprecipitation was followed by FLAG (Usp9x) or HA (Ets-1) immunoblotting. Input lysate was also immunoblotted. Bottom—Map and summary of the Usp9x deletion constructs and their Ets-1 binding activity. The position of the ubiquitin C-terminal hydrolase (UCH) in the catalytic domain is shown by bold letters. Numbers and letters designate highlighted amino acids. **(f)** Immunoblot for Usp9x, Ets-1 and actin in control and Usp9x KD WM1366 NRAS mutant cells treated ± MG132 for 8 h (10 μM). **(g)** HEK293T cells ectopically expressing FLAG-Ets-1 and HA-Ubiquitin were subjected to control or Usp9x KD (left) or treated with vehicle or G9 (2.5 μM, 6 h—right). FLAG immunoprecipitation was followed by HA blotting to detect Ub-Ets-1 levels. Immunoblot for FLAG (Ets-1) in the pulldowns (top) and input lysate (Usp9x and actin—bottom) is shown.

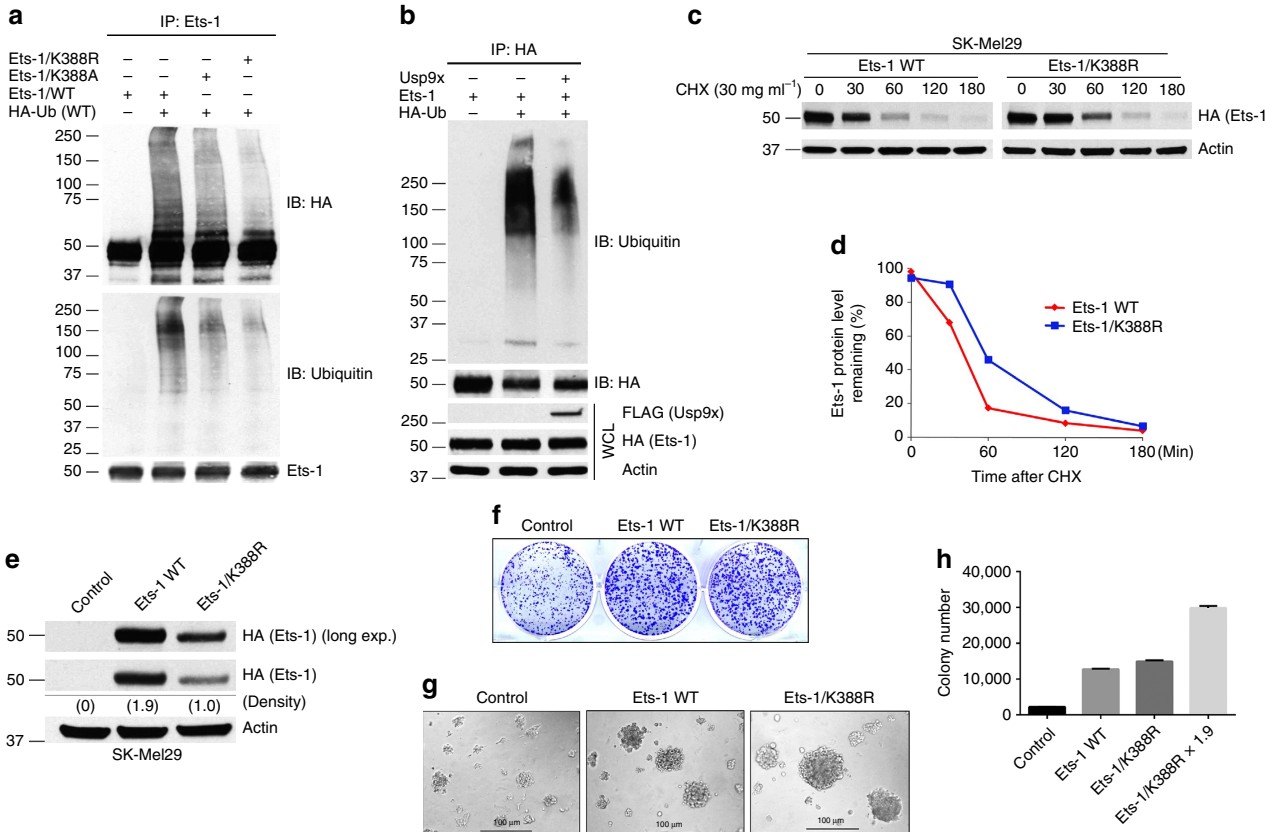

**Figure 4 | Site-specific Ets-1 deubiquitination by Usp9x.** (**a**) HEK293T cells ectopically expressing HA-Ets-1 (WT), HA-Ets-1/K388A or HA-Ets-1/K388R co-expressed with HA-Ub were subjected to immunoprecipitation with Ets-1 antibody (Bethyl) followed by immunoblotting for HA (top) or Ubiquitin (bottom). Ets-1 in the pulldown was also immunoblotted with anti-Ets-1 (Bethyl—bottom). (**b**) HEK293T cells ectopically expressing HA-Ets-1 alone or co-expressed with FLAG-Usp9x and HA-Ub (as noted) were subjected to HA (Ets-1) immunoprecipitation followed by immunoblotting of Ubiquitin (top). Whole cell lysates (WCL) were also immunoblotted for the protein indicated (bottom). (**c**) BRAF mutant SK-Mel29 cells were stably transfected with HA-Ets-1 WT or the K388R mutant plasmid, treated with 30 μg ml$^{-1}$ of cycloheximide (CHX), and harvested at the time points indicated after CHX addition. Immunoblot for HA (Ets-1) is shown. (**d**) The blot from **c** was subjected to densitometric scanning (ImageJ software) to detect changes in HA-Ets-1 protein levels over time. (**e**) Immunoblot for HA and actin in SK-Mel29 cells stably expressing HA-Ets-1 WT or HA-Ets-1/K388R. Protein expression levels were quantified by densitometry (ImageJ software). (**f**) Colony growth (detected by crystal violet staining) of SK-Mel29 cells expressing control, HA-Ets-1 WT or HA-Ets-1/K388R and grown 21 days in standard 2D culture. (**g**) Phase contrast images of SK-Mel29 cells expressing control, HA-Ets-1 WT or HA-Ets-1/K388R and grown on matrigel for 7 days. (**h**) Quantification of growth of colonies in (**f**) after 21 days. All data shown are mean values ± s.d. (error bar) from three replicates.

tumour cells grown in 3D (Supplementary Fig. 3a, right and Supplementary Fig. 3c). Similar effects were noted in both NRAS and BRAF mutant melanoma cells following Ets-1 or Usp9x KD (Supplementary Fig. 3d). Finally, Usp9x KD in ERG-positive prostate cancer cells (VCaP) reduced NRAS protein content (Supplementary Fig. 3e). Thus, Usp9x-mediated stabilization of Ets-1 (and ERG) regulates NRAS expression. To further examine Usp9x regulation of Ets-1 and NRAS expression, Ets-1 and NRAS levels were evaluated in melanoma cell lines with modulated Usp9x expression. Usp9x KD reduced both Ets-1 and NRAS levels, while its overexpression increased both proteins (Fig. 5a). Usp9x KD paradoxically increased pERK levels, suggesting a more complex regulation of the RAS/MEK/ERK pathway by Usp9x. Dusp4 is a phosphatase capable of dephosphorylating ERK and JNK kinases[47,48] and was found to be a potential Usp9x target (Supplementary Data File 1). This was confirmed in pulldown, knockdown and degradation protection assays (Supplementary Fig. 4a–d), and Dusp4 modulation appears to underlie activation of ERK in Usp9x KD cells. However additional studies and analysis of the Usp9x ubiquitylome will be needed to confirm the sufficiency of Usp9x-mediated regulation of Dusp4 levels as an independent mediator of ERK

activation. As expected, either Ets-1 or Usp9x overexpression in SK-Mel29 cells increased 3D tumour growth (Fig. 5b), while Ets-1 KD blocked both control and Usp9x-enhanced 3D growth and colony formation (Fig. 5c,d). Usp9x KD reduced the stability of Ets-1 in both BRAF (Fig. 5e) and NRAS (Fig. 5f) mutant melanoma and decreased NRAS, but not total RAS protein levels. We confirmed regulation of Ets-1/NRAS levels by Usp9x using a doxycycline-inducible Usp9x KD vector (TRIPz) in WM1366 cells (Supplementary Fig. 3f). Both Usp9x and Ets-1 KD consistently and effectively suspended 3D growth of NRAS mutant melanoma (Fig. 5g) derived from metastatic lesions. Overall, Usp9x appears to control ubiquitination of proteins essential in melanoma 3D growth (Ets-1) and attenuation of kinase signalling (Dusp4).

Usp9x, Ets-1 and NRAS protein expression was further assessed in a tissue microarray containing tumour and normal tissue. In normal skin, Usp9x, Ets-1 and NRAS were detected at low levels, with slight accentuation of Ets-1 and NRAS in basal keratinocytes (Fig. 5h, Supplementary Fig. 5a). Benign nevi showed modest staining for Usp9x and minimal staining for NRAS and Ets-1. One nevus expressed higher Usp9x levels in superficial dermal nests in a maturation pattern similar to that

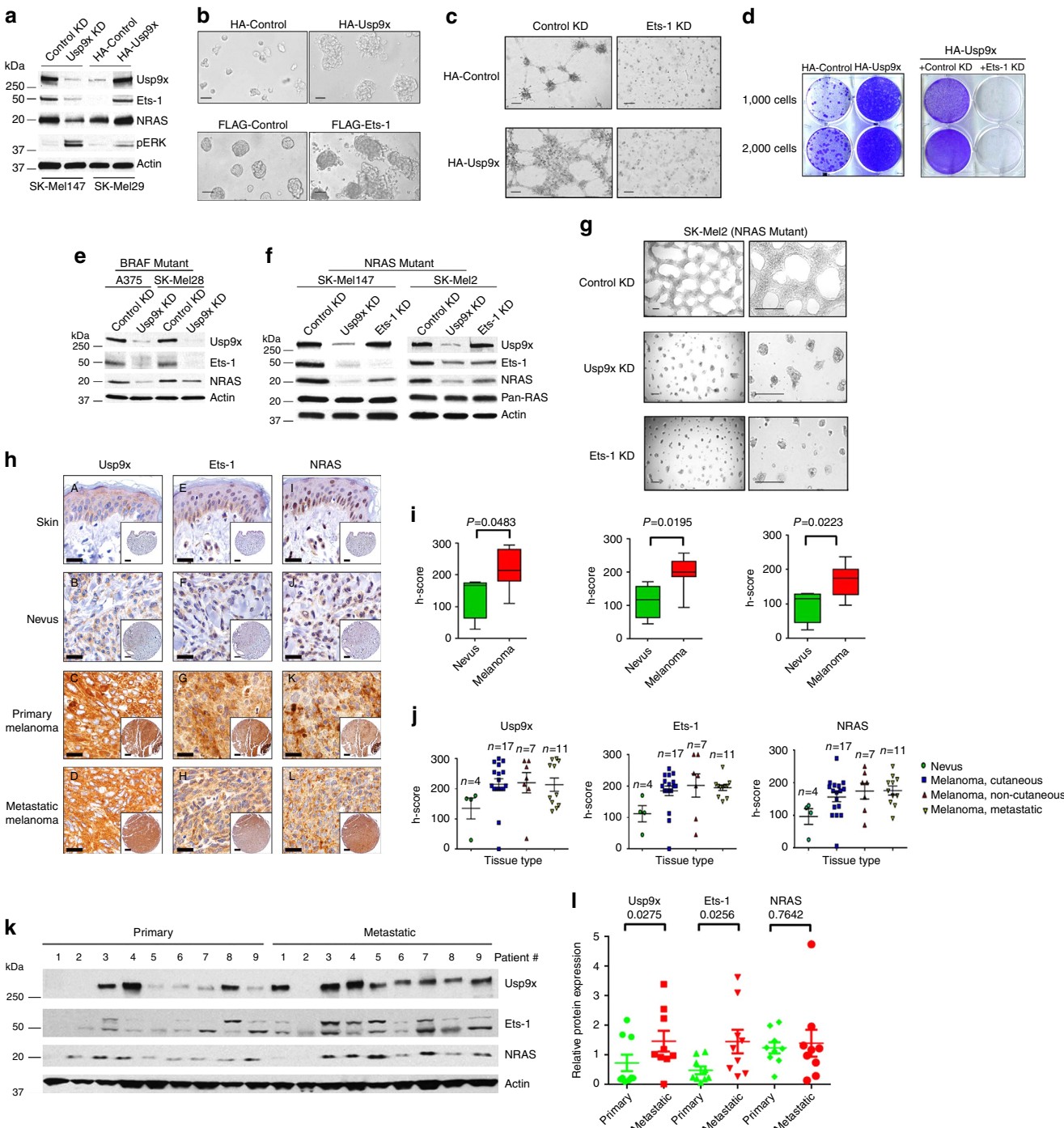

**Figure 5 | Usp9x overexpression in tumours correlates with increased Ets-1 and NRAS protein expression.** (**a**) Immunoblot for Usp9x, Ets-1, NRAS, pERK and actin in control and Usp9x KD SK-Mel147 cells and HA-Control and HA-Usp9x-overexpressing SK-Mel29 cells. (**b**) Phase contrast images of SK-Mel29 cells expressing HA-Control or HA-Usp9x and grown on matrigel for 7 days (top) or SK-Mel29 cells expressing Flag-Control or Flag-Ets-1 and grown on matrigel for 7 days (bottom). Scale bars, 100 µm. (**c**) Phase contrast images of HA-Control and HA-Usp9x expressing SK-Mel29 cells alone or with Ets-1 KD grown on matrigel for 7 days. Scale bars, 100 µm. (**d**) Colony growth (detected by crystal violet staining) of SK-Mel29 cells expressing HA-Control or HA-Usp9x after 21 days in standard 2D culture (left) or after Ets-1 KD before plating (right). (**e**) Immunoblot for Usp9x, Ets-1, NRAS and actin in BRAF mutant cell lines 5 days after Usp9x KD. (**f**) Immunoblot for Usp9x, Ets-1, NRAS, Pan-RAS and actin in NRAS mutant cell lines after 5 days of KD. (**g**) Phase contrast images of NRAS-mutant SK-Mel2 cells with or without Usp9x KD and Ets-1 KD and grown in 3D (matrigel) for 7 days. Scale bars, 500 µm (100 µm inset). (**h**) Immunostaining for Usp9x, Ets-1 and NRAS in normal skin, benign nevi, primary melanoma and metastatic melanoma (insets show whole tissue microarray). Scale bars, 20 µm. (**i,j**) Quantitation of Usp9x, Ets-1 and NRAS immunohistochemical staining by multiplying staining percentage (0–100%) by staining intensity on a numerical scale (none = 1, weak = 2, moderate = 3, strong = 4). (**k**) Immunoblot for Usp9x, Ets-1, NRAS and actin in nine primary and nine metastatic melanoma tumours. (**l**) Quantification of Usp9x, Ets-1 and NRAS expression in immunoblots from nine primary and nine metastatic melanoma patient tumours.

described for HMB45 (refs 49,50), and Usp9x/Ets-1/NRAS staining co-localized in this sample (Supplementary Fig. 5b, yellow versus red arrows). There was co-incident and significant overexpression of Usp9x, Ets-1 and NRAS in melanoma versus nevi (Fig. 5i), but Usp9x expression was not notably different between primary and metastatic melanoma (Fig. 5j; Supplementary Fig. 5a). Analysis of fresh tumour tissue from primary or metastatic sites (Supplementary Table 1) by immunoblotting suggested that Usp9x positivity was more common in metastatic (8/9) than primary tumour (3/9) and correlated with higher Ets-1 (or its isoform) levels in most Usp9x-expressing tumours (Fig. 5k). NRAS levels trended toward higher expression in Ets-1/Usp9x-positive samples, but did not reach statistical significance (Fig. 5l). Melanoma tumours pre-characterized as efficient metastasizers[51] showed higher expression of Usp9x, Ets-1 and NRAS protein than those with inefficient metastatic activity (Supplementary Fig. 5c). Assessment of high-resolution images suggested that Ets-1 was localized in both the cytoplasm and nucleus, particularly in tumour tissues (Supplementary Fig. 5d) as previously noted with other ETS proteins[52,53]. Altogether, these results suggest that Usp9x overexpression is an early event in expansion of primary and metastatic melanoma, involving stabilization of Ets-1 to amplify NRAS expression.

**Usp9x stabilizes Ets-1 to induce NRAS expression.** To define a mechanism for regulation of NRAS expression by Usp9x in melanoma, we examined the effect of Usp9x (or Ets-1) on NRAS promoter activity. Previous ChIP-SEQ studies in other cell lines (Supplementary Fig. 6) confirmed multiple ETS sites in the NRAS promoter region. We cloned the NRAS promoter from SK-Mel147 cells and established a luciferase reporter construct. In two melanoma cell lines (SK-Mel29, WM1366; Fig. 6a,b), Usp9x activated NRAS promoter activity by ∼2-fold, while Ets-1 expression increased promoter activity by >2.5-fold. ChIP-SEQ defined 5 ETS sites (designated *E1M* through *E5M*) on the NRAS promoter (Fig. 6c), which were individually mutated to define their involvement in ETS responsiveness. *E1M, E2M, E3M* and *E4M* point mutations suppressed ETS promoter activity (Fig. 6d), suggesting cooperation between sites. Mutation of *E5M* had minimal effect. To assess the effect of Usp9x knockdown on Ets-1 levels on chromatin, chromatin-immunoprecipitation of Ets-1 and NRAS promoter PCR were performed (ChIP-PCR) on nuclear extracts from control and Usp9x KD WM1366 cells. Usp9x KD markedly reduced the recovery of Ets-1 bound to the NRAS promoter (Fig. 6e). Thus, Ets-1 appears to mediate NRAS expression by binding multiple sites in the NRAS promoter and is subject to regulation by Usp9x.

**Usp9x is a valid tumour target in melanoma.** In addition to their role in tumorigenicity and NRAS regulation, Usp9x and Ets-1 may control responsiveness to kinase inhibition. We noted constitutive overexpression of nuclear Ets-1 in a melanoma cell model of vemurafenib resistance[54] and previously reported that G9 overcame this resistance via DUB inhibition[39] (Supplementary Fig. 7a–c). Recent publications have described downregulation of several ETS family proteins following kinase inhibition, but specific upregulation of Ets-1 has been noted in cells treated with a BRAF inhibitor (Supplementary Fig. 7d–f), suggesting a distinct regulatory mechanism exists for Ets-1 (refs 55–57). Short-term inhibition of MEK or BRAF kinase activity with small molecules (PD 0325901, vemurafenib) blocked ERK activation but increased Ets-1 and NRAS expression in BRAF-mutant SK-Mel29 cells (Supplementary Fig. 7g), suggesting that MEK inhibition reverses a negative feedback

loop suppressing Ets-1 expression[55,56]. We confirmed that both MEK- (PD) and BRAF- (vemurafenib) inhibition increased Ets-1 gene and protein expression in a time-dependent fashion (Fig. 7a–e) and also increased NRAS promoter activity (Fig. 7f). Usp9x KD blocked kinase inhibitor-induced Ets-1 and NRAS expression (Fig. 7g) and correlated with greater cell growth inhibition (Fig. 7h) and apoptosis (Fig. 7i) than that activated by kinase inhibition alone. Ets-1 KD caused similar changes in cells treated with kinase inhibitor (Fig. 7j).

To determine whether Usp9x-targeting agents could have clinical value in melanoma patients, we evaluated G9 activity in an *in vivo* model of NRAS mutant melanoma. G9 rapidly reduced Ets-1 protein levels in NRAS mutant cells (Fig. 8a). Mice inoculated with NRAS mutant SK-Mel147 cells were treated with G9, PD or their combination, and tumour growth was assessed over a 3-week treatment interval. Both G9 and PD reduced tumour growth (Fig. 8b), but tumour cells refractory to either agent began to emerge by the end of the treatment interval (Fig. 8b, right). Combined G9 and PD treatment completely blocked tumour growth measured *in vivo*, (Fig. 8b, right) which was confirmed by end of study assessment of tumour weight (Fig. 8c) and appearance (Fig. 8d). To further assess the clinical potential of DUB inhibition in melanoma therapy, tumour derived from a patient with NRAS mutant melanoma (M405—Supplementary Fig. 5c) was established in NSG mice and treated with vehicle or G9. G9 treatment blocked tumour growth, assessed by tumour volume (Fig. 8e) and end of study tumour size (Fig. 8f) and weight (Fig. 8g) measurements. In addition, Ets-1 protein levels were significantly reduced in tumours from G9-treated mice (Fig. 8h,i). These results suggest that DUB inhibition can suppress tumour growth and enhance the antitumor activity of kinase inhibitors by reducing Ets-1 protein content and NRAS expression in melanoma.

**Discussion**

Usp9x has been shown to be overexpressed or mutated in several cancers, but its effects on tumorigenesis have been difficult to define, possibly because of the context-specific function of its many substrates[17]. We noted that melanoma was unexpectedly dependent on Usp9x for 3D growth and *in vivo* expansion, with potential Usp9x addiction noted in NRAS mutant melanoma. We found that Usp9x KD or inhibition induced major changes in the melanoma ubiquitylome when assessed by ubiquitin-remnant enrichment, suggesting that modification of multiple proteins could underlie the observed effects of Usp9x on melanoma. However, each potential modification needs to be validated as Ub-peptide sequence information alone does not fully discriminate between 'hits' and true or effector substrates, as noted with specific members of the ETS family (Fig. 3c) in this study. Within this hit list, we identified Ets-1 as a Usp9x substrate and key mediator of Usp9x dependence in melanoma. We further demonstrated that Ets-1 promotes NRAS gene expression, which may at least partly underlie the high sensitivity of melanoma to Usp9x inhibition and Ets-1 depletion. Since NRAS mutations occur in a broad range of tumour types[38], those regulated by Ets-1 (or other member of the ETS family) may be treatable through Usp9x inhibition. Indeed, previous reports have shown Usp9x deubiquitinates and stabilizes ERG, and our previously described DUB inhibitor (WP1130) demonstrated anti-tumour efficacy in ERG-driven prostate cancer[15]. The Usp9x-deubiquitation site on Ets-1 (K388) shares sequence identity with previously defined sites of interaction between ETS proteins and Usp9x, suggesting that Usp9x may stabilize other ETS family members (ERG, FLI1, FEV) through this specific recognition motif (MNY(D/E)K*LSR)[15]. Additional studies are needed to confirm this. It is worth noting that

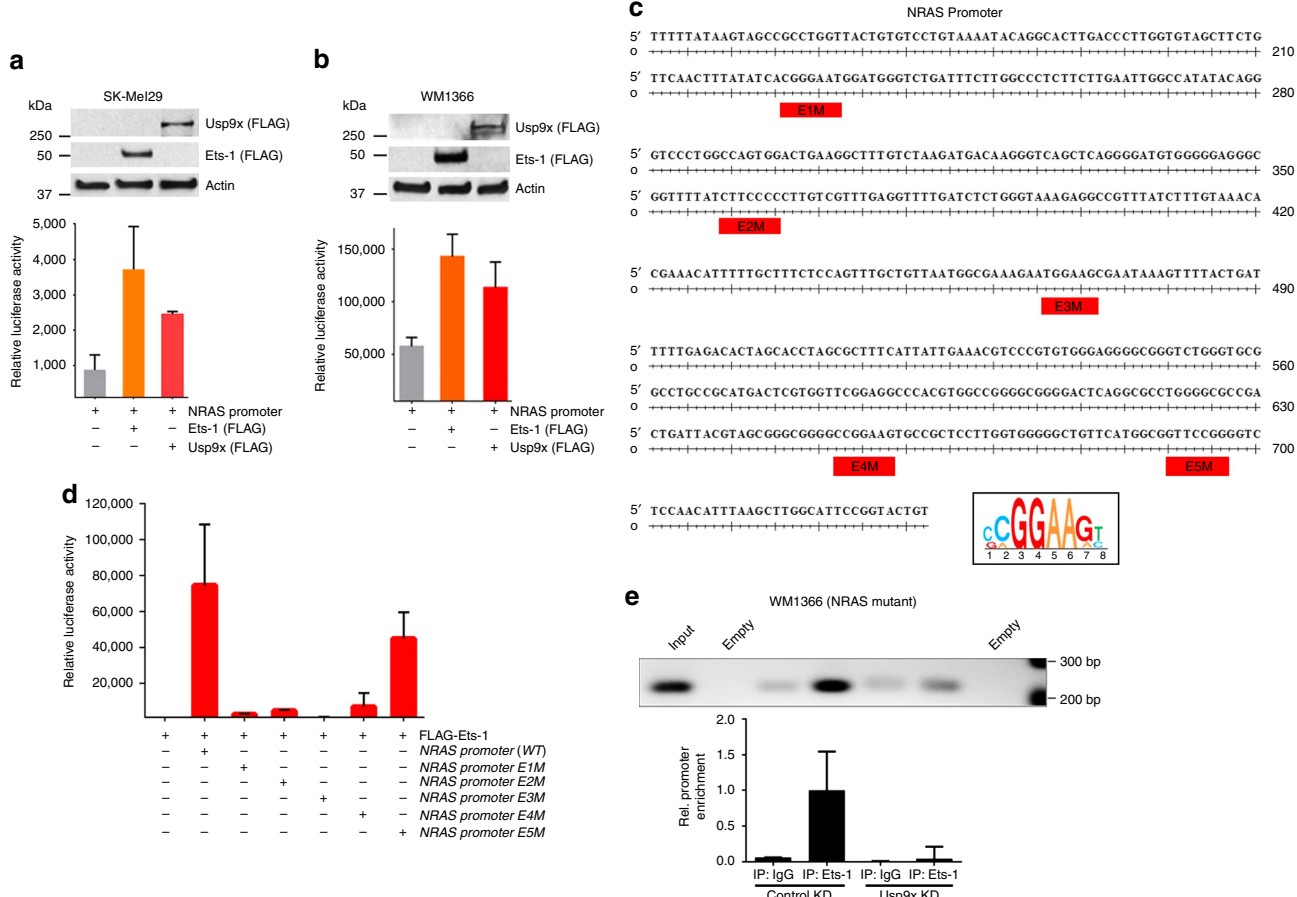

**Figure 6 | Ets-1 activates the proximal NRAS promoter.** (**a**) Immunoblot for FLAG in BRAF mutant SK-Mel29 cells (express low endogenous Usp9x and Ets-1 levels) stably transfected with FLAG-Usp9x or FLAG-Ets-1 (top). Relative luciferase units (firefly/Renilla) in lysates from SK-Mel29 cells expressing (48 h) the proximal NRAS promoter, FLAG-Ets-1 or FLAG-Usp9x (bottom). (**b**) Immunoblot for FLAG in NRAS mutant WM1366 cells expressing FLAG-Ets-1 or FLAG-Usp9x (top). Relative luciferase units (firefly/Renilla) in lysates from WM1366 cells expressing the proximal NRAS promoter, FLAG-Ets-1 or full-length FLAG-Usp9x (bottom). (**c**) Proximal NRAS promoter sequence cloned from NRAS mutant SK-Mel147 cells, highlighting 5 putative ETS sites (designated E1M through E5M) derived from ChIP-SEQ analysis in other cell lines and visual inspection of the sequence. The consensus ETS binding sequence is highlighted below (boxed). (**d**) Relative luciferase units (firefly/Renilla) in lysates from SK-Mel29 cells expressing FLAG-Ets-1 and the proximal NRAS promoter (WT) or point mutants of each ETS putative binding site in the promoter region (E1M, E2M, E3M, E4M and E5M). (**e**) DNA-protein crosslinks from control and Usp9x KD cells were subjected to immunoprecipitation (as noted) before being used to prime a PCR reaction to detect the NRAS promoter. PCR products are shown (top) and compared with the input fraction (unfractionated DNA–protein complexes). Relative enrichment of the NRAS promoter for each condition is graphed below and represents the ave. ± s.d. of three independent experiments.

non-mutant NRAS is also transcriptionally activated by Ets-1 and controllable by Usp9x. Thus, tumours dependent on elevated wild-type NRAS expression (for example, basal-like breast cancer)[58] may also be highly responsive to Usp9x inhibition. Other RAS regulatory proteins were also detected in the Usp9x ubiquitylome (that is, RIN, RSU1)[59,60] and may contribute to the effects of Usp9x inhibition on the NRAS pathway. However, regulation of specific ETS proteins by Usp9x may also have implications outside the NRAS regulatory network. For example, ETS proteins can bind to mutated upstream promoters of critical genes (that is, hTERT) and may also underlie the biological importance of Usp9x in melanoma and other tumours[30,31].

Analysis of the Usp9x ubiquitylome predicted a diverse group of substrates, including a number of targets within the UPS, but whether these are valid targets or are regulated directly or indirectly by Usp9x requires further investigation. As we recently noted, inactivation of Usp9x leads to expression of a closely related enzyme (Usp24) as a compensatory mechanism[43]. To account for dynamic changes caused by Usp9x KD, we compared the ubiquitylome generated after Usp9x KD to that induced by

our recently characterized DUB inhibitor with activity against Usp9x (ref. 43). About 40% of targets were common to both conditions, including some previously defined by other approaches (Supplementary Data File 6). One common target, Ets-1, was pursued based on its biologic role in tumour expansion and involvement in the RAS/MEK/ERK pathway. Dusp4 was selected based on similar criterion. The ubiquitylomes generated with G9 and Usp9x KD probably had incomplete overlap because G9 targets other DUBs, including Usp24 and Usp5 (refs 39,43). UbiScan analysis did not capture all previously defined Usp9x targets, perhaps because of limitations of the technique or differences in gene expression in the cell type examined here. In addition, protein ubiquitination and turnover may have kinetics that cannot be fully resolved by single time point studies and knockdowns performed in one cell line. Definitive identification of substrates for Usp9x and other UPS proteins in specific tissues will require a combination of genetic and biochemical approaches.

Our studies indicate that Usp9x may be a good therapeutic target in melanoma because of its effects on tumour expansion,

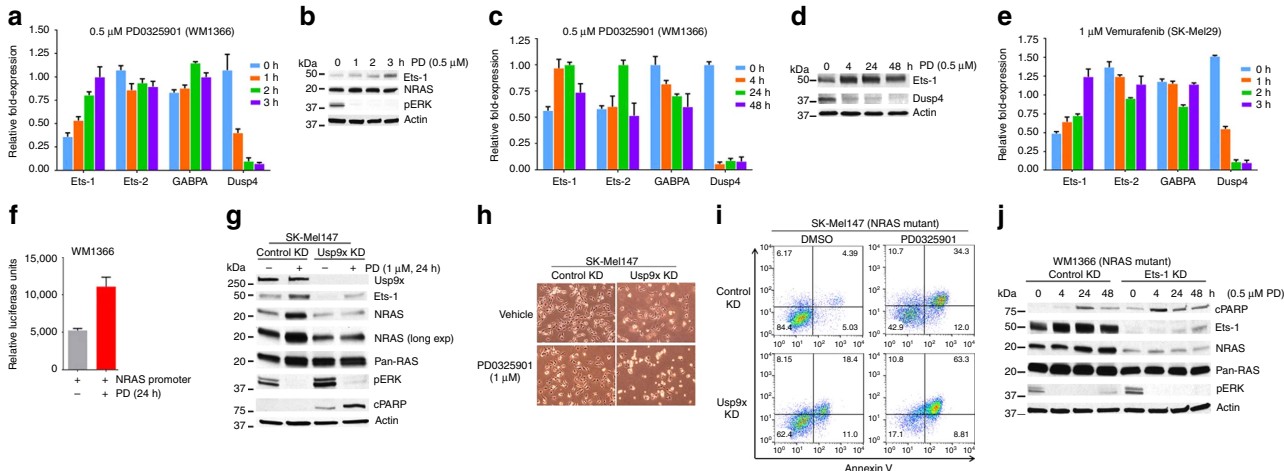

**Figure 7 | Ets-1 expression induced by BRAF and MEK inhibitors is blocked by Usp9x inhibition.** (**a**) Expression levels of the indicated genes (Ets-1, Ets-2, GABPA and Dusp4) by RT-PCR in NRAS mutant (WM1366) cells treated with PD0325901 for 0–3 h. (**b**) Immunoblot of the indicated proteins in NRAS mutant (WM1366) cells treated with PD0325901 for the interval noted. (**c**) Expression levels of the indicated genes (Ets-1, Ets-2, GABPA and Dusp4) by RT-PCR in NRAS mutant (WM1366) cells treated with PD0325901 for the interval noted. (**d**) Immunoblot for the proteins indicated in NRAS mutant (WM1366) cells treated with PD0325901 as described. (**e**) Expression levels of the genes indicated (Ets-1, Ets-2, GABPA and Dusp4) by RT-PCR in BRAF mutant (SK-Mel29) cells treated with vemurafenib for the interval indicated. (**f**) Relative luciferase units (firefly/*Renilla*) from NRAS mutant (WM1366) cells expressing the NRAS promoter for 24 h and treated with PD0325901 (0.5 μM) as noted. (**g**) Immunoblot for the proteins indicated in control and Usp9x KD NRAS mutant (SK-Mel147) cells treated with PD0325901 as indicated. (**h**) Phase contrast images of control and Usp9x KD NRAS mutant (SK-Mel147) cells treated with PD0325901 for 48 h. (**i**) Annexin V assessment in control and Usp9x KD NRAS mutant (SK-MEL147) cells treated with PD0325901 (1 μM) for 48 h as indicated. (**j**) Immunoblot for the proteins indicated in control and Ets-1 KD NRAS mutant (WM1366) cells treated with PD0325901 as indicated.

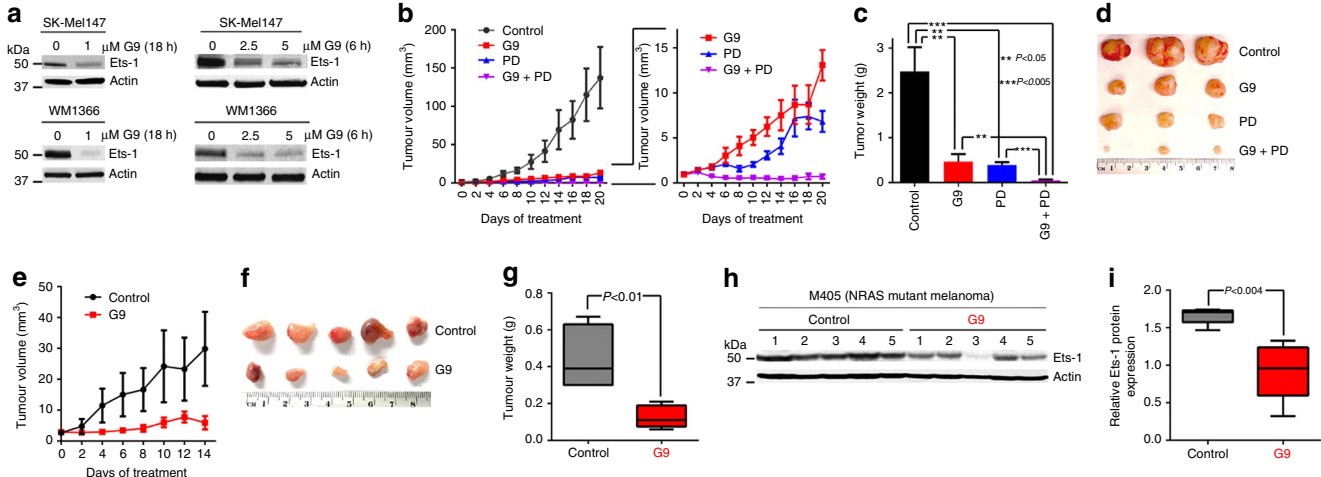

**Figure 8 | Usp9x inhibition has anti-melanoma activity.** (**a**) Immunoblot for Ets-1 in NRAS mutant SK-Mel147 (top) or WM1366 (bottom) cells treated with G9 (1 μM) for the interval and condition indicated. Actin was blotted as a loading control (**b**) Left—Tumour volumes in NSG mice injected subcutaneously with SK-Mel147 cells and treated intraperitoneally with either vehicle, G9 (15 mg kg$^{-1}$, QOD), PD0325901 (5 mg kg$^{-1}$; OD) or both for 3 weeks (N = 3/group). Right—Comparison of tumour growth in inhibitor treated mice. (**c**) Average ± s.d. of tumour weight (from **b**) at the end of treatment (Day 21). (**d**) Photographs of individual tumours (from **b**) at the end of treatment. (**e**) Tumour volumes in NSG mice injected subcutaneously with tumour derived from a patient with NRAS mutant melanoma (M405) and treated intraperitoneally with either vehicle or G9 (15 mg kg$^{-1}$, QOD) for 2 weeks (N = 5/group). (**f**) Photographs of individual tumours (from **e**) at the end of treatment. (**g**) Average ± s.d. of tumour weight at the end of treatment (from **e**, day 14). (**h**) Immunoblot for assessment of Ets-1 protein levels in tumours from (**e**) Actin was blotted as a loading control. (**i**) Ets-1 protein levels (from **h**) were quantified by densitometry (ImageJ software).

regulation of Ets-1 stability, NRAS expression and response to kinase inhibitors. However, other Usp9x substrates may also add (for example, Mcl-1) or diminish (for example, Dusp4) anti-tumour activity of Usp9x inhibition and will need to be further examined in melanoma and other tumours. In melanoma, both MEK and BRAF inhibition led to an induction of Ets-1 and NRAS expression that could be blocked by Usp9x inhibition. Combined kinase and DUB inhibition was effective in completely suppressing NRAS-mutant melanoma *in vivo*, suggesting combination therapy may prevent resistance mediated by Ets-1 induction. Usp9x inhibition is expected to add to the treatment options for patients with Ets-1-overexpressing tumours, particularly when used in rational, biologically based combinations. Equally attractive, Usp9x inhibition may be an effective means of targeting NRAS-mutant and -dependent tumours, a goal that has been particularly elusive with other approaches.

## Methods

**Cell culture.** A375, SK-Mel2, WM1366 (ATCC), SK-Mel28, SK-Mel29, SK-Mel147 and SK-Mel103 cell lines were provided by Dr Monique Verhaegen (University of Michigan, Ann Arbor, MI, USA). The A375R (vemurafenib-resistant) cell line was a kind gift from Dr Juxiang Cao (Boston University School of Medicine, Boston, Massachusetts, USA). HEK293T cells were primarily maintained in Dulbecco's Modified Eagle's Medium (DMEM). The VCaP2 (prostate) cell line was provided by Dr Arul Chinnaiyan (University of Michigan, Ann Arbor, MI, USA) and cells were cultured in DMEM Glutamax. All media was supplemented with 10% heat-inactivated FBS (Atlanta Biological), 2 mM L-glutamine and 1% penicillin/streptomycin (GIBCO). Tet Free FBS was from Omega Scientific, Inc.

**Antibodies.** Primary antibodies used in this study include: NRAS, Pan-RAS, HRAS, Dusp4, Ubiquitin (total), Ets-2, GABPAα, Mcl-1, HA, β-Actin, ERK2 (total) (Santa Cruz); KRAS (Calbiochem, OP24); AF-6, Usp9x, Ets-1 (Bethyl Laboratories); pERK, Caspase8, PARP, BID, BIM (Cell Signaling); ERG (Abcam); HA (Roche); FLAG (Sigma). Blots were developed with ECL substrate (Pierce) and imaged on X-ray film (BioExpress). Antibody catalogue numbers and their dilutions are included in Supplementary Table 2.

**Three-dimensional cultures (3D).** Equal numbers of viable control, KD and overexpressing cells from each cell type (1,000 cells per well or as indicated) were grown on growth factor-reduced Matrigel (Catalogue # 354230; BD transduction) for 7 days[61]. Phase contrast images were acquired at ×5 or ×10 resolution on a Leica inverted microscope. For cells treated with small molecule inhibitors, media was exchanged every 3 days. To quantify the number of colonies, total numbers of colonies from 2 to 3 wells of an 8-well chamber slide were counted using phase contrast images acquired at ×5 resolution. For spheroid culture, $10^6$ cells were plated in complete media on 100 mm dishes coated with 1% agarose. The cells were allowed to grow for 2–3 days. The spheroids were collected, pelleted, lysed in lysis buffer and subjected to immunoblot analysis.

**Assessment of the Usp9x ubiquitylome (UbiScan).** *Sample preparation and mass spectrometry.* Cells were collected in Urea Lysis Buffer (20 mM HEPES (pH 8.0), 9.0 M urea, 1 mM sodium orthovanadate (activated), 2.5 mM sodium pyrophosphate, 1 mM ß-glycerol-phosphate) and processed by Cell Signaling Technology using the Ubiquitin Branch Motif Antibody (CST cat. #3925)[45,46] for PTMScan analysis. Lysates were sonicated, centrifuged at 20,000 g for 15 min and 'cleared' protein extracts were reduced (with DTT), carboxamidomethylated (with iodoacetamide) and normalized for total protein before tryptic digestion (Worthington, cat. #LS003740). Peptides were enriched by solid-phase extraction with Sep-Pak C18 classic cartridges (Waters cat. #WAT051910), lyophilized and re-dissolved. Slurries of the Ubiquitin Branch Motif Antibody were used to recover ubiquitin-remnant peptides, which were eluted from antibody-resin with 0.15% trifluoroacetic acid (100 µl total volume). Peptides were desalted on Empore $C_{18}$ (Sigma) packed tips and eluted with 40% acetonitrile in 0.1% TFA, then loaded directly onto a 10 cm × 75 µm PicoFrit capillary column packed with Magic C18 AQ reverse-phase resin. The column was developed with a linear gradient of acetonitrile in 0.125% formic acid, delivered at 280 nl min$^{-1}$ over a 90-min interval. Analytical replicates were generated by running duplicate samples to increase the number of MS/MS identifications from each sample. A LTQ-Orbitrap Velos mass spectrometer running Xcalibur 2.0.7 SP1 was used to collect tandem mass spectra by the top 20 method, a dynamic exclusion repeat count of 1, and repeat duration of 30 s. A singly charged polysiloxane ion $m/z = 371.101237$ was used for real time recalibration of mass error. SEQUEST and the Core platform from Harvard University were used to evaluate MS/MS spectra and files were searched against the NCBI *Homo sapiens* FASTA Database updated on 27 June 2011 containing 34,899 forward and 34,899 reverse sequences. Precursor ion mass accuracy of ± 5 p.p.m., and 1 Da for product ions was allowed. Protease specificity was limited to trypsin, with at least one tryptic (K- or R-containing) terminus required per peptide and a maximum of four mis-cleavages. Methionine residue oxidation and the di-glycine (K-GG) remnant was allowed on lysine residues and cysteine carboxamidomethylation was specified as a static modification. False discovery rates were estimated using reverse decoy databases and filtered using a 5% FDR in the Linear Discriminant module of Core. We also filtered for the presence of the K-GG motif in peptides.

*Label-free quantitation.* All quantitative results were generated using Progenesis V4.1 (Waters Corporation) or XCalibur 2.0.7 SP1 to extract the integrated peak area of the corresponding peptide assignments according to previously published protocols[45,46]. The Progenesis software incorporates a chromatographic alignment (or time warping) algorithm that performs multiple binary comparisons to generate an overall clustering strategy for the complete data set of all identified peptides on the basis of mass precision. Extracted ion chromatograms for peptide ions that changed in abundance between samples were manually reviewed to ensure accurate quantitation either in Progenesis or using XCalibur software (version 2.0.7 SP1, Thermo Scientific). This eliminated the possibility that the automated process selected the wrong chromatographic peak from which to derive the corresponding intensity measurement. Peak areas were normalized using a log2 median normalization strategy in Progenesis[45,46].

**shRNA-mediated gene knockdown.** Melanoma cells were infected with the lentiviral expression system for short hairpin RNA (shRNA) against human pLVX-Usp9x, kindly provided by Dr Dzwokai Ma (University of California, Santa Barbara)[40]. For NRAS and control KD: pGIPZ Control, pGIPZ-NRAS-1, and pGIPZ-NRAS-2 were obtained from Open Biosystems. Open Biosystems TRIPZ control (clone ID: RHS4743) and TRPIZ human Usp9x (clone ID: V3THS320834) doxycycline-inducible shRNA vectors were also used in melanoma cells. Doxycycline at 1 µg ml$^{-1}$ was used to induce shRNA expression.

Ets-1 shRNA was kindly provided by coauthor, Dr Peter C. Hollenhorst (Indiana University, Bloomington, Indiana). HEK293T cells were transfected with the lentiviral packaging vectors pMD2.G and psPax2 (Addgene) together with the shRNA vectors to produce virus using PolyFect as described by the manufacturer (QIAGEN). The medium was changed to DMEM with 10% fetal bovine serum, and after 48 h, viral supernatant was collected. Viral supernatant containing 4 µg ml$^{-1}$ of Polybrene (Sigma-Aldrich) was added to each melanoma cell line. Cells with stable KD were selected with puromycin.

**Chemical reagents.** EOAI3402143 (referred to as G9) was synthesized and provided by Cheminpharma (Branford, CT). Other reagents used in this study were obtained from the following sources: hemagglutinin-tagged ubiquitin vinyl methyl sulfone (HA-UbVS; Boston Biochem); vemurafenib (PLX4032; Chemie Tek); PD 0325901 (Cayman Chemical). All reagents were made up and stored frozen as 10 mM stock solutions.

**Crystal violet colony staining.** Equal numbers of viable SK-Mel29 (or A375) cells with modified gene expression were grown in 6-well plates for 3 weeks and subjected to crystal violet staining (3.7% paraformaldehyde (PFA), 0.05% Crystal Violet in distilled water (filter at 0.45 um)) for 20 min at room temperature. The plate was photographed by scanning.

**DUB-labelling assays.** To assay DUB activity, melanoma cells were lysed in DUB buffer (50 mM Tris pH 7.2, 5 mM MgCl$_2$, 250 mM sucrose, protease inhibitor cocktail (Roche), 1 mM NaF and fresh 1 mM PMSF) for 10 min at 4 °C followed by brief sonication. The lysates were centrifuged at 20,000 g for 10 min, and the supernatants (20 µg) were incubated with 2 µM of HA-UbVS for 75 min at 37 °C, followed by boiling in reducing sample buffer and resolving by SDS–polyacrylamide gel electrophoresis (SDS–PAGE). DUBs were detected by HA immunoblotting[62].

**Lysate preparation and western blotting.** Total cell lysates were prepared by sonicating and boiling cell pellets in × 1 Laemmli-reducing sample buffer. Detergent-soluble cell lysates were prepared by lysing cells in cold isotonic lysis buffer (10 mM Tris–HCl, pH 7.5, 0.1% Triton X-100, 150 mM NaCl, protease inhibitor cocktail and 1 mM PMSF) for 15 min on ice and centrifuging for 10 min at 20,000 g. The clarified supernatant was used as the detergent-soluble cell fraction. Primary and metastatic melanoma tumours were isolated from patients, and a small portion was sliced, minced and snap frozen with liquid nitrogen followed by homogenization in lysis buffer. Lysates were electrophoresed (SDS–PAGE gels) and transferred to nitrocellulose membranes (Whatmann). Proteins were detected by immunoblotting. Uncropped western blots of key figures are presented in Supplementary Fig. 8.

**Plasmids.** For overexpressing Usp9x, p3xFlag-Usp9x was created by 3-way cloning using PCR to amplify a 320 bp N-terminal fragment of Usp9x with the StuI site in Usp9x. Forward: 5′- tgtacgaagcttacagccacgactcgtggctc-3′; Reverse: 5′ggaaccacccatcgaggcc-3′. The PCR product was cut with HindIII and StuI. pCDNA5-TAP-Usp9x was cut with StuI and NotI. These fragments were ligated into p3XFlag-CMV10 (Sigma) linearized with HindIII/NotI. A PCR was performed with forward primer 5′-gctctagatctatgggactacaaagacc-3′ and the reverse primer described above. This product was cut with BglII and StuI, and ligated together with the StuI/BamHI fragment from p3XFlag-Usp9x together with MIGR1 linearized with BglII. pcDNA3-Usp9x-HA was kindly provided by Dr Dzwokai Ma (University of California, Santa Barbara)[40]. 3xFlag-Ets-1 and pGL4.25 were kindly provided by Dr Peter C. Hollenhorst (Indiana University, Bloomington, Indiana)[24]. HA-Ets-1 (WT) was kindly provided by William G. Kaelin, Jr. (Dana-Farber Cancer Institute, Boston). Approximately, 5 µg of each pCDNA3 and pCDNA3-Usp9x-HA plasmid, and 2 µg of p3xFlag-Ets-1 WT, HA-Ets-1WT and HA-Ets-1/K388R were used for overexpression in SK-Mel29 and A375 cells.

**MTT assay.** Cells were seeded in a 96-well plate at 5,000 per well in the presence of the indicated concentration of compound for 3 days in a $CO_2$ incubator at 37 °C. Twenty microliters of 5 g l$^{-1}$ MTT solution was added to each well for 2 h at 37 °C. The cells were then lysed in 10% SDS buffer, and absorbance at 570 nm relative to a reference wavelength of 630 nm was determined with a microplate reader. To examine proliferation using the MTT assay, cells were plated in triplicate and processed for MTT assay as described above.

**Quantitative RT-PCR.** Melanoma cells were grown on 100 mm dishes with or without PD 0325901 or vemurafenib for 0–48 h followed by RNA isolation using the RNeasy kit (Qiagen, Valencia, CA). Samples for qRT-PCR were prepared with $\times 1$ SYBR Green PCR Master Mix (Applied Biosystems, Foster City, CA) and primers listed in Supplementary Information. The primers were optimized for amplification under the following reaction conditions: denaturing at 95 °C for 10 min, followed by 40 cycles of 95 °C for 15 s and 60 °C for 1 min. Melting curves were analysed for all samples after completion of the amplification protocol. GAPDH was used as the housekeeping gene for control expression. All RT-PCR primers were purchased from RealTimePrimers.com.

**Analysis of Ets-1 ubiquitination in 293T cells.** HEK293T cells grown in DMEM with 10% FBS were co-transfected with Flag-Ets-1 and HA-ubiquitin expression plasmids. For analysis of the effects of Usp9x KD, the cells were transfected with shRNAs against Usp9x or a non-targeting shRNA for 72 h before plasmid transfection. For the analysis of the effects of G9, cells were co-transfected with Flag-Ets-1 and HA-ubiquitin (WT) expression vectors for 40 h, then treated with G9 (2.5 µM) for 5 h. Cells were lysed in 1% NP-40, 1% SDS, 2 mM EDTA, 1 mM NEM (fresh) and 25 mM Tris–HCl, pH 7.5, boiled for 20 min, and then diluted with 10 volumes of immunoprecipitation buffer (lysis buffer with 1% NP-40). Lysate of 500 µg was immunoprecipitated with anti-FLAG overnight and then with 30 µl protein A/G for 2 h at 4 °C. The beads were washed five times with immunoprecipitation buffer and 0.1 M NaCl. Western blot analysis was performed with anti-HA or ubiquitin antibody to detect ubiquitinated Ets-1. Usp9x, FLAG and actin were probed by immunoblotting.

**Immunoprecipitation for K63-linked ubiquitination.** To assess ubiquitination of Ets-1, HEK293T cells were co-transfected with FLAG-Ets-1, pRK5-HA-ubiquitin (WT), pRK5-HA-Ub/K48 only or pRK5-HA-Ub/K63 only (obtained from Dr Vaibhav Kapuria (University of Lausanne, Switzerland)), and after 48 h, cells were lysed in 1% NP-40, 1% SDS, 2 mM EDTA, 1 mM NEM (fresh) and 10 mM Tris–HCl, pH 7.5. Lyses were boiled for 20 min and then diluted with 10 volumes of immunoprecipitation buffer (lysis buffer with 1% NP-40). FLAG was immunoprecipitated as described above. Western blot analysis was performed with anti-HA or ubiquitin antibody to detect ubiquitinated Ets-1.

**Immunoprecipitation for Ets-1/K388 mutant ubiquitination.** The Ets-1/K388 (K388A, K388R) mutant was generated using a Quickchange II Site-Directed mutagenesis Kit on the HA-Ets-1 construct (Agilent Technologies). Primer sets used in mutagenesis are provided in Supplementary Table 3. To assess ubiquitination of Ets-1, HEK293T cells were co-transfected with HA-Ets-1, HA-Ets-1/K388A or HA-Ets-1/K388R with pRK5-HA-ubiquitin (WT), and immunoprecipitation with Ets-1 antibody (Bethyl, Montgomery, TX) was performed as described above. Western blot analysis was performed with the anti-HA and ubiquitin antibody.

**Usp9x and Ets-1 immunoprecipitation.** For immunoprecipitation of endogenous Usp9x, SK-Mel2 cells were lysed in lysis buffer (25 mM HEPES (pH 7.5), 400 mM NaCl, 0.5% IGEPAL CA-630, 5% glycerol, protease inhibitors and 1 mM fresh PMSF). The soluble fraction of the lysate (1 mg) was diluted to adjust NaCl and IGEPAL CA-630 concentrations to 100 mM and 0.125%, respectively. Precleared lysates were incubated with a rabbit control IgG or anti-Usp9x antibody (5 µg) (Bethyl) at 4 °C for 3 h with rotation, followed by immunoprecipitation with Protein A/G PLUS Agarose (Santa Cruz Biotechnolgy) beads at 4 °C for 1 h with rotation. Beads were washed five times with 100 mM NaCl and 0.1% IGEPAL CA-630 and boiled in Laemmli buffer for Western blot analysis. Anti-Ets-1 was used to immunoprecipitate Ets-1 as described above.

**Co-immunoprecipitation for Usp9x and Ets-1.** FLAG-Usp9x WT, FLAG-Usp9x-CDM, FLAG-Usp9x E1, FLAG-Usp9x E1M, FLAG-Usp9x E5 (ref. 43) and HA-Ets-1 WT plasmids were transfected into HEK293T cells. Forty-eight hours after transfection, cells were lysed in lysis buffer (25 mM HEPES (pH 7.5), 400 mM NaCl, 0.5% IGEPAL CA-630, 1 mM NEM (fresh) 1 mM DTT, 5% glycerol and protease inhibitors) and the soluble fraction of the lysate was diluted to adjust NaCl and IGEPAL CA-630 concentrations to 100 mM and 0.125%, respectively. Lysate of 0.5 mg was immunoprecipitated with anti-HA (Ets-1) overnight and then with 40 µl protein A/G for 2 h at 4 °C. Beads were washed five times with 100 mM NaCl and 0.1% IGEPAL CA-630, and boiled in Laemmli buffer for Western blot analysis. Western blot analysis was performed with anti-FLAG antibody (Usp9x).

**Apoptosis measurement.** An Annexin V-fluorescein isothiocyanate (FITC) staining assay was performed as previously described[43]. The cells were seeded in six-well plates and exposed to compounds as indicated for 48 h. The cells were then trypsinized, washed with cold PBS, and stained with Annexin V-FITC for 10 min on ice. Positive cells were detected by flow cytometry.

**Xenograft studies.** NSG (NOD/SCID/IL2r-g (null)) mice were injected mid-dorsally with $3 \times 10^5$ BRAF mutant SK-Mel29 expressing HA-control, HA-Usp9x cells, or $5 \times 10^5$ NRAS mutant SK-Mel 147 cells in 0.1 ml of Matrigel/DMEM suspension. $5 \times 10^5$ M405 (NRAS mutant) patient-derived melanoma tumour cells[51] in 0.1 ml of Matrigel/L15 suspension were also inoculated in NSG mice. Tumours were allowed to reach about 10 mm$^3$, after which mice were tumour-size matched and assigned to treatment groups consisting of vehicle, PD 0325901 or G9 as indicated. G9 and PD 0325901 were administered in DMSO: PEG300 (1:1) by i.p. injection every other day at 15 mg kg$^{-1}$ for G9 and every day for PD 0325901 at 5 mg kg$^{-1}$. Tumour size was monitored by calipers every other day using the following formula: volume $= (\text{width})^2 \times \text{length} \times \text{height}/2$. Animal weight was also recorded every other day.

**Tissue banking.** The tissue bank protocol used for this study was developed and approved jointly by the clinical director of the University of Michigan (UM) melanoma program, UM Cancer Center director of tissue procurement, UM chief of anatomic pathology, and UM director of the section of dermatopathology. The protocol was developed to avoid any compromise in patient care, pathologic diagnosis, tumour staging, or treatment. Patient confidentiality was maintained by password and firewall-protected access to all pertinent databases. Melanoma specimens were obtained with informed consent from all patients according to protocols approved by the Institutional Review Board of UM Medical School (HUM00102527). All patients included in this study had stage II or III melanoma proven by biopsy (most often needle core). A small (typically 2–6 mm) tissue sample was obtained from surgically resected tumours. Most of the melanomas in this study were regional stage III lymph node or skin/soft tissue disease with palpable, clinically enlarged node(s) or soft tissues.

**Luciferase assays.** Luciferase assays used a Dual Luciferase Reporter Assay System (Promega) according to the manufacturer instructions. NRAS promoter sequences (733 bp) from the NRAS mutant melanoma cell line SK-Mel147 were cloned upstream of the firefly luciferase-pGL3-Basic (Promega) plasmid cut with Hind III and XhoI (Forward Primer- 5′-AGACTCGAGGAGGAGTGCC-3′ -XhoI, Reverse Primer- 5′-GATCAAGCTTAAATGTTGGAGACCCCGGAA-3′—HindIII) and site-directed mutagenesis of promoter and expression constructs were performed using the Quickchange Lightning Multi Site-Directed mutagenesis Kit (Agilent Technologies). Primer sets used in mutagenesis are provided in Supplementary Table 4. Positive clones were confirmed by the UM sequence core. Melanoma SK-Mel29 and WM1366 cells were plated at ~50% confluence in a 6-well plate ($3 \times 10^5$ cells per well) 24 h before transfection. Cells were transfected with 1 µg of each p3xFlag, p3xFlag-Usp9x, p3xFlag-Ets-1, HA-Ets-1, wild-type (WT) or point mutant NRAS promoter constructs (WT, E1M, E2M, E3M, E4M and E5M), 2 µg of firefly and 200 ng of Renilla plasmid using PolyFect Transfection Reagent (Qiagen). After 48 h, media was removed, and cells were washed two times with $\times 1$ PBS, resuspended in 500 µl $\times 1$ PLB, disrupted by one freeze/thaw cycle ($-80$ °C) and dissociated with a BD 1 ml 26G syringe. Luciferase activity was measured in 20 µl of cell lysate using a BD PharMingen (Monolight 3010C) luminometer. Firefly values were normalized to Renilla values.

**Melanoma tissue microarray (TMA) immunohistochemistry.** For immuno-histochemical analysis, tissue microarrays (TMA) were used, containing 36 cases of melanoma and 12 cases of normal and non-melanoma tumour tissues of the skin in duplicates (96 cores) (Catalogue No.: Z7020108, BioChain Institute, Inc.). All the tissues were from surgical resection. They were fixed in 10% neutral-buffered formalin for 24 h. Ninety-four cores consisted of 8 normal skin, 8 benign nevi, 4 cases of non-melanoma skin cancer basal cell carcinoma (BCC), 4 cases non-melanoma skin cancer squamous cell carcinoma (SCC), 48 malignant melanomas and 24 metastatic melanomas. Immunohistochemistry was also performed on individual slides for Usp9x, Ets-1 and NRAS. Formalin-fixed, paraffin sections were cut at 5 microns and rehydrated to water. Heat-induced epitope retrieval was performed with FLEX TRS high pH retrieval buffer (9.01) for 20 min. After per-oxidase blocking, the antibody was applied at room temperature for 60 min. The FLEX HRP EnVision System was used for detection. DAB chromagen was then applied for 10 min. Slides were counterstained with Harris Hematoxylin for 5 s and then dehydrated and coverslipped. Tumour content of each core or slide was verified by H&E staining. Immunohistochemistry was performed using anti-Usp9x (1:1,000, Abcam), Ets-1 (1:500, Bethyl) and NRAS (1:150, Origene, clone 5G7). Slides were scored by a UM dermatologist and pathologist (Dr Paul William Harms) for percentage of positive cells and intensity of staining. All positive cases displayed nuclear and cytoplasmic staining. Photomicrographs were taken with a SPOT Insight Colour camera (Diagnostic Instruments) on an Olympus BX41 microscope with Olympus UPlanFL $\times 10$ and $\times 40$, $\times 200$ and $\times 400$ objectives using SPOT Basic software.

**Immunofluorescence.** BRAF mutant A375 parental and vemurafenib-resistant cells were grown on 6-well slides for 24 h. Media was then decanted and the wells were washed $3 \times$ with PBS. Cells were fixed in methanol for 20 min at $-20$ °C, washed $3 \times$ with PBS and blocked for 1 h at room temperature in PBS with 0.3% Tween-20 and 5% BSA. The primary antibody was Ets-1 (Bethyl), which was

diluted 1:100 in PBS with 0.3% Tween-20 and 1% BSA (antibody dilution buffer) and incubated overnight at 4 °C. After 3 × washes in PBS, Alexa Fluor 488 anti-Rabbit secondary antibody (Life Technologies) was added at 1:500 in antibody dilution buffer and with DAPI incubated for 1 h at room temperature. After 5 × washes in PBS, slides were coverslipped with ProLong Gold anti-fade reagent (Invitrogen). Images were acquired using an Olympus Fluo View 500. A representative image from each sample is shown.

**Chromatin immunoprecipitation assay (ChIP).** WM1366 cells were seeded at a density of $5 \times 10^7$ in 150 mm dishes and protein/DNA cross-linking was induced with formaldehyde at 1% final concentration at room temperature for 15 min. Crosslinking was terminated by the addition of 1/10 volume 1.25 M glycine for 5 min at room temperature followed by cell lysis (1% SDS, 10 mM EDTA, 50 mM Tris, pH 8) for 10 min and sonication (Misonix, Microson Ultrasonic Cell Disruptor (20 s on, 40 off, 10 amplitude for 30 min) resulting in an average chromatin fragment size of 300 bp. DNA–protein complexes were immunoprecipitated with 5 μg of rabbit Ets-1 (Bethyl) or 5 μg rabbit IgG antibody (Santa Cruz) overnight at 4 °C (1:10 ml volume) in dilution buffer; (20 mM Tris at pH 8, 2 mM EDTA, 150 mM NaCl, 0.01% Triton X-100, 0.01% SDS, protease inhibitors) and rotated overnight and then with 50 μl Dynabeads for 2 h at 4 °C. Beads were washed with low salt (0.1% SDS, 1% Triton X-100, 2 mM EDTA, 20 mM Tris at pH 7.5, 150 mM NaCl), high salt (0.1% SDS, 1% Triton X-100, 2 mM EDTA, 20 mM Tris at pH 7.5, 500 mM NaCl), LiCl (250 mM LiCl, 1% NP-40, 1% deoxycholic acid, 1 mM EDTA, 10 mM Tris at pH 7.5) and TE wash buffer (10 mM Tris, pH 7.5, 1 mM EDTA) twice. Beads were resuspended with 250 μl of fresh elution buffer (1% SDS, 50 mM NaHCO₃). Elutes were resuspended with 5 M NaCl (10 μl in 250 μl of elute) and incubated at 65 °C overnight, followed by 10 μg ml⁻¹ RNase A addition and incubated for 30 min at 37 °C. ChIP DNA was purified using a Quick PCR Purification Kit (Qiagen). Primers used for NRAS promoter detection (244 bp): forward primer (5-GTAGCCGCCTGGTTACTG-3), reverse primer (5-CCCAGAGATCAAAACCTC-3). RT-PCR was performed as described above.

**Statistical analysis.** All statistical analysis was carried out using GraphPad Prism software (GraphPad Prism 6 and GraphPad InStat3). For quantitative data, treatment groups were reported as mean ± s.d. and compared using the unpaired Student's *t*-test. Usp9x, Ets-1 and NRAS expression values were categorized into low/moderate ($<$ 300 product score) and high ($>$ 300 product score). Statistical significance was established at $P \leq 0.05$ unless otherwise noted. Data points are shown as the mean ± s.d.

**Institutional approval.** Protocols utilizing animals were reviewed and approved by the University Animal Care and Use Committee (University of Michigan). All patient samples were obtained through signed informed consent using a protocol reviewed and approved by the Institutional Review Board (University of Michigan).

**Data availability.** Mass spectrometry proteomics data have been deposited with the ProteomeXchange Consortium via the PRIDE partner repository with the data set identifier PXD005417 (ref. 63). Access details include: Website: http://www.ebi.ac.uk/pride, Project name: Ubiquitin remnant analysis in melanoma post inhibition of Usp9x. Project accession: PXD005417. Project DOI: Not applicable. G9 may be made available through a materials transfer agreement (MTA). All other remaining data are available within the Article and its Supplementary Files, or available from the authors on request.

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

## Acknowledgements

We thank Jessica Mercer for editing the manuscript and Jeffrey Silva and Matthew Stokes (Cell Signaling Technology Inc.) for analysis and discussion of the Usp9x ubiquitylome. We also thank Arul Chinnaiyan, Monique Verhaegen (University of Michigan, Ann Arbor, MI, USA) for kindly providing cell lines and William G. Kaelin, Jr (Dana-Farber Cancer Institute, Boston) for kindly providing Ets-1 HA plasmid. Vaibhav Kapuria (University of Lausanne, Switzerland) kindly provided pRK5-HA-Ub WT, K48 and K68 plasmids and Yihong Liu provided technical assistance with this study. We also thank Pinki Chowdhury (University of Michigan, Ann Arbor, MI, USA) for kindly providing technical assistance for ChIP analysis. We thank Nisha Meireles, Clinical Research Specialist, Multidisciplinary Cutaneous Oncology Program, for database and data management. We thank the ENCODE project consortium and Richard Myers for the use of data sets. We also would like to thank the patients who agreed to be part of an IRB-approved translational study. We acknowledge support from the Allen H. Blondy Research Fund for Melanoma (to M.T., H.P.), The Harry J. Lloyd Charitable Trust and the Michigan Translational Research and Commercialization (MTRAC) program (to N.J.D.).

## Author contributions

H.P., L.F.P., M.K., A.P. and H.S. performed the research and analysed the data. P.W.H. analysed TMA and protein expression in clinical samples. P.C.H., U.E., A.D. and M.T. contributed materials. H.P. and N.J.D. designed the study, analysed the data and wrote the manuscript. All authors contributed to data review and provided comments on the manuscript.

## Additional information

**Competing financial interests:** A patent (Patent No: US 8,809,377 B2, Date of Patent: Aug. 19, 2014) covering the synthesis and use of G9 has been filed with L.F.P., M.T. and N.J.D. as authors and constitutes a competing financial interest. The remaining authors declare no competing financial interests.

