## [Peer Review File · Nature Communications]

Reviewers' comments:

Reviewer #1 (Remarks to the Author): expert in melanoma

In this manuscript, Potu and colleagues examine the effects of ETS1 regulation by the deubiquitinase Usp9X, which deubiquitinates ETS1, and stabilizes it. They show that changes in ETS affect metastatic progression of cells. They also show that ETS1 binds to NRAS promoter, and that ETS1 is increased during the acquisition of resistance to BRAF inhibitor treatment, in turn increasing NRAS activity. By blocking USP9X, and allowing for degradation of ETS1, the investigators were able to overcome BRAF inhibitor resistance in preclinical models. The manuscript is well written, and the conclusions, for the most part, are supported by the data. Overall this is a strong study, identifying the contribution of ubiquitination of ETS1 and effects on NRAS leading to BRAFi resistance.

Some concerns below should be addressed:

Figure 2C: How many mice were used in 2c? 2e? N for all experiments should be included.

Figure 3A. The legend on the top of the graph is illegible. In fact, overall the figures were very inconsistent in size, not well organized in terms of space, illegible in places, and hard to read. This detracts significantly from some otherwise very nice work.

In figure 5, while the overall expression of ETS1 is increasing, it appears to be excluded from the nucleus in the tissue microarray. If the rationale is that ETS1 is increasing, binding to the NRAS promoter, and driving resistance, one would expect ETS1 to be in the nucleus. Can the authors provide an explanation for this? Perhaps higher resolution images would help clarify this.

In figure 7: Since NRAS is not changing in the NRAS mutant cells upon Mek inhibition, how do the authors explain the connection between ETS1, NRAS and resistance in these cells? I note that there is an NRAS promoter activity assay in d- is this the rationale for the NRAS activity downstream of the Mek inhibitor? If so why is this run at 24 hours, and the others run at a maximum of 3 hours? What do the gene profiles look like at 24 hours?

Also the graph in Figure 7A is mislabeled as SK29 cells, when I believe they are WM1366 cells.

In figure 7i, the flow pictures are illegible- zooming results in low resolution, and the numbers on the graph cannot be read.

In the experiments examining the effects of Vemurafenib on ETS1 and other genes in SKMel29, there are no blots showing whether NRAS, or NRAS activity is increased upon Mek inhibitor treatment.

In the experiments in Figure 8, the difference between the G9, PD or combo are so small as to be negligible. However, it is possible that the tumors take off and grow eventually in the single agent treatments, but not in the combo. Have the authors left those experiments for longer?

Reviewer #2 (Remarks to the Author): expert in melanoma therapy resistance

A. Knockdown or pharmacologic inhibition of the deubiquitinating enzyme Usp9x blocks proteasomal destruction of numerous proteins including the transcription factor Ets-1. The data presented suggests that Ets-1 activity/stability plays a vital role in NRAS expression and tumorigenicity particularly in NRAS driven melanomas. Suppressing Usp9x expression/function leads to a reduction in Ets-1

culminating in reduced 3D growth and xenograft growth especially in combination with BRAF/MEK inhibition.

B. This is a novel finding that enhances our understanding of NRAS driven melanoma biology and establishes a role for Usp9x and Ets-1 in melanoma and may lead to clinical evaluation. While multiple human melanoma cell lines were used in this study (including several NRAS Q61 mutants), delineating the mechanism(s) of Usp9x/Ets-1 is a difficult proposition given the large number of additional mutations each cell line surely possesses. As such many of the differences of expression were quite subtle.

C. Data quality and presentation were outstanding. Overall the methods used were appropriate.

D. Statistical analysis and use of controls was adequate.

E. Based on data presented, the conclusions drawn in this robust study are reasonable.

F. Suggested improvements: Figure 1a, Casp 8 and Bid are shown with slight cleaved Casp 8 and t-Bid present primarily in the BRAF A375 cells. Given the previously defined role of Usp9x in Mcl-1 preservation and enhanced apoptosis resistance, evaluation of Bim expression is more relevant than Bid since Bim has been shown to be a key Mcl-1 antagonist and may also be a target of Usp9x. Bid on the other hand is cleaved by Casp-8 in response to extrinsic and some intrinsic apoptotic cascades.

Figure 2a, subtle pERK upregulation - should have total ERK as well (pERK could be related to increased ERK expression via ETs-1?).

Figure 3g/5f, Knockdown of Usp9x/Ets-1 is not nearly complete throughout this study, which is a weak point overall. CRISPR or even inducible shRNA (e.g., pTRIPZ) would be superior to stable shRNA clones.

Figure 7g/i/j, Apoptosis is promoted in Usp9x KD NRAS melanoma cells treated with MEKi - showed Mcl-1 and Bid on an earlier untreated blot - would be nice to see them again along with Bim.

G. References appear appropriate.

H. No problems with clarity and context throughout this manuscript

Reviewer #3 (Remarks to the Author): Expert in de-ubiquitinating enzymes and cancer

The manuscript by Potu et al. presents evidence suggesting a role for the deubiquitinating enzyme Usp9x in the regulation of Ets-1 protein levels, which in turn impact the tumorigenic program of metastatic melanoma. The authors claim that Usp9x overexpression is an early event in primary and metastatic melanoma, involving stabilization of Ets-1 and amplification of NRAS expression. The study also provides details on how stabilization of Ets-1 directly regulates transcription of NRAS. In addition, the authors emphasize that combination therapy (DUB and kinase inhibition) is effective in suppressing NRAS-mutant melanoma in vivo. Overall this is an interesting piece of work. The manuscript is well written, clearly presented, and in general the data nicely match the conclusions.

The in vivo data on the effect of Usp9x in tumor expansion and the possibility to inhibit tumor growth by blocking or inhibiting Usp9x are very compelling. However, the analysis of the tissue microarray containing tumor and normal tissues does not provide convincing evidence that the Usp9x-Ets-1-NRAS axis is the crucial mechanism in play for establishment and metastasis of the tumor.

Specific comments about how authors might improve the manuscript are described below.

1) an examination of the chromatin environment upon depletion of Usp9x to demonstrate that, indeed, the increase in NRAS expression is mainly controlled by Ets-1 levels on the promoter DNA and is independent of the chromatin state of the gene.

2) an explanation of results obtained with vemurafenib treatment; how do the authors reconcile the increase in Ets-1 and NRAS expression upon inhibition of MEK or BRAF kinase activity and the effect the Usp9x KD has on this increase.

3) The authors state that they detect a paradoxical increase in ERK activation in Usp9x KD cells.

However, poor attempts are made to dissect the pathway for other possible targets of Usp9x. A more detailed explanation of this part would improve the manuscript and put their findings into a broader perspective.

Minor comments:

1) The introduction should include more detail of the connection between ERK kinase pathway and regulation of the ETS class of transcription factors.

2) It has been previously reported that Usp9x deubiquitinates the anti-apoptotic protein MCL-1. However, the authors only observe a decrease in MCL-1 levels in two out of the three melanoma cell lines present in the manuscript (Figure 1a).

3) Figure 1f: the appearance of the Ub-linked form of Usp9x is not accompanied by a decrease in protein levels of the unmodified DUB.

4) A better explanation of the increase in pERK levels observed in Figure 2a upon Usp9x KD.

5) In Figure 3c the decrease in Ets-1 levels upon Usp9x depletion does not correlate with an increase in NRAS levels.

6) Misspelling of Ets-1 in Figure 4e-g.

7) Normalization parameters used in Figure 4h do not allow for a direct comparison as they imply linearity in the type of response, whereas the authors suggest that a signal expansion mechanism is involved in tumorigenesis.

8) In Figure 4e the authors show that Ets-1 K388R levels are lower than the wt counterpart. This difference is recovered upon overexpression in 293T cells. No comment on this discordancy is present in the manuscript.

Response to Reviewers Comments. (*Authors responses are shown in bold italics*).

Reviewer #1: expert in melanoma

In this manuscript, Potu and colleagues examine the effects of ETS1 regulation by the deubiquitinase Usp9X, which deubiquitinates ETS1, and stabilizes it. They show that changes in ETS affect metastatic progression of cells. They also show that ETS1 binds to NRAS promoter, and that ETS1 is increased during the acquisition of resistance to BRAF inhibitor treatment, in turn increasing NRAS activity. By blocking USP9X, and allowing for degradation of ETS1, the investigators were able to overcome BRAF inhibitor resistance in preclinical models. The manuscript is well written, and the conclusions, for the most part, are supported by the data. Overall this is a strong study, identifying the contribution of ubiquitination of ETS1 and effects on NRAS leading to BRAFi resistance.

Some concerns below should be addressed:

Figure 2C: How many mice were used in 2c? 2e? N for all experiments should be included.

Au: These numbers are now included for all animal studies.

Figure 3A. The legend on the top of the graph is illegible. In fact, overall the figures were very inconsistent in size, not well organized in terms of space, illegible in places, and hard to read. This detracts significantly from some otherwise very nice work.

Au: Thanks for pointing that out. We've corrected that figure and others that were illegible or hard to read.

In figure 5, while the overall expression of ETS1 is increasing, it appears to be excluded from the nucleus in the tissue microarray. If the rationale is that ETS1 is increasing, binding to the NRAS promoter, and driving resistance, one would expect ETS1 to be in the nucleus. Can the authors provide an explanation for this? Perhaps higher resolution images would help clarify this.

Au: We have now included additional and supporting analysis of Ets-1 expression and activity in the nucleus. First, we added high resolution images in Supplementary Fig. 5d that suggest Ets-1 expression in the nucleus as well as the cytoplasm of melanoma cells. In addition, we have now added CHIP-PCR analysis of Ets-1 binding to the NRAS promoter in control and Usp9x KD cells in Fig. 6e, which suggests that Ets-1 occupancy at the NRAS promoter site is detectable in control cells and diminished by Usp9x KD. These data add to the already existing evidence for nuclear function of Ets-1 from promoter-reporter studies and tissue microarray analysis.

In figure 7: Since NRAS is not changing in the NRAS mutant cells upon Mek inhibition, how do the authors explain the connection between ETS1, NRAS and resistance in these cells? I note that there is an NRAS promoter activity assay in d- is this the rationale for the NRAS activity downstream of the Mek inhibitor? If so why is this run at 24 hours, and the others run at a maximum of 3 hours? What do the gene profiles look like at 24 hours?

Au: NRAS is changing with Mek inhibition (see Fig. 7g and j, Supplementary Fig.7g) but requires Ets-1 induction to drive NRAS expression. We used short and longer term incubation intervals with Mek inhibitor to demonstrate rapid Ets-1 induction (at the gene and protein level) and typically see peak NRAS induction after 24 to 48 hours. Together, the data suggests that Mek inhibition induces Ets-1 expression to subsequently enhance NRAS expression by Ets-1 occupancy at the NRAS promoter (Fig. 6d,e).

Also the graph in Figure 7A is mislabeled as SK29 cells, when I believe they are WM1366 cells.

Au: Thanks for pointing that out. This has been corrected.

In figure 7i, the flow pictures are illegible- zooming results in low resolution, and the numbers on the graph cannot be read.

Au: We are sorry for the low resolution and have now corrected that figure.

In the experiments examining the effects of Vemurafenib on ETS1 and other genes in SKMel29, there are no blots showing whether NRAS, or NRAS activity is increased upon Mek inhibitor treatment.

Au: That data is shown in Supplementary Fig.7g.

In the experiments in Figure 8, the difference between the G9, PD or combo are so small as to be negligible. However, it is possible that the tumors take off and grow eventually in the single agent treatments, but not in the combo. Have the authors left those experiments for longer?

Au: Statistically the reviewer is correct, there is no difference between G9 or PD treatment and tumor size compared to controls (Fig. 8c). However, there is a significant difference between single treatment regimens and their combination, supporting the reviewers point regarding outgrowth from single agent treatment versus their combination. Therefore, since we did see a significant effect of combo vs. single agent regimens, we did not further extend the treatment interval.

Reviewer #2 (Remarks to the Author): expert in melanoma therapy resistance

A. Knockdown or pharmacologic inhibition of the deubiquitinating enzyme Usp9x blocks proteasomal destruction of numerous proteins including the transcription factor Ets-1. The data presented suggests that Ets-1 activity/stability plays a vital role in NRAS expression and tumorigenicity particularly in NRAS driven melanomas. Suppressing Usp9x expression/function leads to a reduction in Ets-1 culminating in reduced 3D growth and xenograft growth especially in combination with BRAF/MEK inhibition.

B. This is a novel finding that enhances our understanding of NRAS driven melanoma biology and establishes a role for Usp9x and Ets-1 in melanoma and may lead to clinical evaluation. While multiple human melanoma cell lines were used in this study (including several NRAS Q61 mutants), delineating the mechanism(s) of Usp9x/Ets-1 is a difficult proposition given the large number of additional mutations each cell line surely possesses. As such many of the differences of expression were quite subtle.

Au: Thank you for that assessment. We are certainly aware of the other genetic defects in these cells but note similar responses in growth and Ets-1 regulation in all after Usp9x KD.

C. Data quality and presentation were outstanding. Overall the methods used were appropriate.

D. Statistical analysis and use of controls was adequate.

E. Based on data presented, the conclusions drawn in this robust study are reasonable.

F. Suggested improvements: Figure 1a, Casp 8 and Bid are shown with slight cleaved Casp 8 and t-Bid present primarily in the BRAF A375 cells. Given the previously defined role of Usp9x in Mcl-1 preservation and enhanced apoptosis resistance, evaluation of Bim expression is more relevant than Bid since Bim has been shown to be a key Mcl-1 antagonist and may also be a target of Usp9x. Bid on the other hand is cleaved by Casp-8 in response to extrinsic and some intrinsic apoptotic cascades.

Au: Thanks for that suggested improvement. We assessed Bim isoforms in those same cells (Fig.1) and did not see consistent effects on these cells in response to Usp9x KD.

Figure 2a, subtle pERK upregulation - should have total ERK as well (pERK could be related to increased ERK expression via Ets-1?).

Au: We now include blots for total ERK in Fig. 2a, which did not show changes in response to Usp9x KD.

Figure 3g/5f, Knockdown of Usp9x/Ets-1 is not nearly complete throughout this study, which is a weak point overall. CRISPR or even inducible shRNA (e.g., pTRIPZ) would be superior to stable shRNA clones.

Au: Thank you for that great suggestion. We used TRIPZ vectors and doxycycline induction to show similar effects of Usp9x KD on Ets-1 and NRAS as seen in stable shRNA clones (Supplementary Fig. 3f).

Figure 7g/i/j, Apoptosis is promoted in Usp9x KD NRAS melanoma cells treated with MEKi - showed Mcl-1 and Bid on an earlier untreated blot - would be nice to see them again along with Bim.

Au: This was not pursued since we did not see major impact of Usp9x KD on those proteins in earlier studies.

G. References appear appropriate.

H. No problems with clarity and context throughout this manuscript

Reviewer #3 (Remarks to the Author): Expert in de-ubiquitinating enzymes and cancer

The manuscript by Potu et al. presents evidence suggesting a role for the deubiquitinating enzyme Usp9x in the regulation of Ets-1 protein levels, which in turn impact the tumorigenic program of metastatic melanoma. The authors claim that Usp9x overexpression is an early event in primary and metastatic melanoma, involving stabilization of Ets-1 and amplification of NRAS expression. The study also provides details on how stabilization of Ets-1 directly regulates transcription of NRAS. In addition, the authors emphasize that combination therapy (DUB and kinase inhibition) is effective in suppressing NRAS-mutant melanoma in vivo. Overall this is an interesting piece of work. The manuscript is well written, clearly presented, and in general the data nicely match the conclusions.

The in vivo data on the effect of Usp9x in tumor expansion and the possibility to inhibit tumor growth by blocking or inhibiting Usp9x are very compelling. However, the analysis of the tissue microarray containing tumor and normal tissues does not provide convincing evidence that the Usp9x-Ets-1-NRAS axis is the crucial mechanism in play for establishment and metastasis of the tumor.

Au: Thanks for that assessment. We have added new data to strengthen the basis for existence of a Usp9x-Ets-1-NRAS axis, as noted in the response to reviewers 1 and 2. Specifically, we've added new data, including Fig. 6e, and Supplementary Figs. 3f and 5d, which further add support for the existence of this signaling axis.

Specific comments about how authors might improve the manuscript are described below.

1) an examination of the chromatin environment upon depletion of Usp9x to demonstrate that, indeed, the increase in NRAS expression is mainly controlled by Ets-1 levels on the promoter DNA and is independent of the chromatin state of the gene.

Au: Thanks for that great suggestion. ChIP data are now included to support the existence of NRAS promoter occupancy by Ets-1 and its control by Usp9x in Fig. 6e.

2) an explanation of results obtained with vemurafenib treatment; how do the authors reconcile the increase in Ets-1 and NRAS expression upon inhibition of MEK or BRAF kinase activity and the effect the Usp9x KD has on this increase.

Au: Ets-1 induction in vemurafenib treated melanoma cells was previously noted by at least two other independent labs (reproduced in Supplementary Fig.7 d-f), and we confirmed that induction in the present study. We suggest that Usp9x acts to deubiquitinate Ets-1 to protect it from destruction and combined with vemurafenib, leads to an induction and accumulation of Ets-1 to drive NRAS expression. Usp9x KD can blunt or block that induction. We are examining possible mediators of Ets-1 induction in response to vemurafenib treatment but have not yet completed that study.

3) The authors state that they detect a paradoxical increase in ERK activation in Usp9x KD cells. However, poor attempts are made to dissect the pathway for other possible targets of Usp9x. A more detailed explanation of this part would improve the manuscript and put their findings into a broader perspective.

Au: There are several possible contributors to ERK activation in Usp9x KD cells and we focused on one key contributor; Dusp4. We provided more detailed discussion of the possible role of this phosphatase in that response (page 10), but other Usp9x substrates may also play a role. This is another major work that, when completed, will be the focus of an additional and separate manuscript.

Minor comments:

1) The introduction should include more detail of the connection between ERK kinase pathway and regulation of the ETS class of transcription factors.

Au: Thanks for that suggestion which has been added to the text on page 4.

2) It has been previously reported that Usp9x deubiquitinates the anti-apoptotic protein MCL-1. However, the authors only observe a decrease in MCL-1 levels in two out of the three melanoma cell lines present in the manuscript (Figure 1a).

Au: We do not see that Mcl-1 is a consistent Usp9x modulated protein, particularly in melanoma cells. This may be due to the requirement for a specific phosphodegron that fortifies recognition by Usp9x that is not present in melanoma but existent in other cell types. Additional studies are needed to draw firm conclusions but is not the focus of the present study.

3) Figure 1f: the appearance of the Ub-linked form of Usp9x is not accompanied by a decrease in protein levels of the unmodified DUB.

Au: It is difficult to see with a high MW protein like Usp9x, which is 290kDa. We have previously shown that the ubiquitin linked and unlinked 100 kD Usp5 shift can be seen (see reference; Oncotarget Potu et.al, 2014).

4) A better explanation of the increase in pERK levels observed in Figure 2a upon Usp9x KD.

Au: We propose that Dusp4 downregulation by Usp9x KD underlies this response in later discussion and figures (Supplementary Fig. 4).

5) In Figure 3c the decrease in Ets-1 levels upon Usp9x depletion does not correlate with an increase in NRAS levels.

Au: NRAS levels should decrease, not increase in response to Usp9x KD. This is shown in the figure.

6) Misspelling of Ets-1 in Figure 4e-g.

Au: Corrected. Thanks for pointing it out.

7) Normalization parameters used in Figure 4h do not allow for a direct comparison as they imply linearity in the type of response, whereas the authors suggest that a signal expansion mechanism is involved in tumorigenesis.

Au: We agree that linearity may not be aligned with the normalization parameter but it was one way of attempting to estimate distinctions in response to proteins with unequal expression.

8) In Figure 4e the authors show that Ets-1 K388R levels are lower than the wt counterpart. This difference is recovered upon overexpression in 293T cells. No comment on this discordancy is present in the manuscript.

Au: We have added a comment to address that discordancy (page 9) where expression of distinct ligases and E2's may underlie this difference in expression of the mutant protein in different cell types.

REVIEWERS' COMMENTS:

Reviewer #1 (Remarks to the Author):

The authors have addressed all concerns in my previous review. I congratulate them on a very nice piece of work.

Reviewer #2 (Remarks to the Author):

The authors adequately addressed the concerns I previously raised and I accept this improved manuscript as is.

Reviewer #3 (Remarks to the Author):

I recommend acceptance. The authors have satisfactorily addressed all my concerns.